# Effective Backdoor Defense by Exploiting Sensitivity of Poisoned Samples

**Weixin Chen[1], Baoyuan Wu[2] ∗, Haoqian Wang[1]∗**
[1]Tsinghua Shenzhen International Graduate School, Tsinghua University
[2]School of Data Science, Shenzhen Research Institute of Big Data,
The Chinese University of Hong Kong, Shenzhen
chenwx20@mails.tsinghua.edu.cn, wubaoyuan@cuhk.edu.cn,
wanghaoqian@tsinghua.edu.cn

## Abstract

Poisoning-based backdoor attacks are serious threat for training deep models on data from untrustworthy sources. Given a backdoored model, we observe that the feature representations of poisoned samples with trigger are more sensitive to transformations than those of clean samples. It inspires us to design a simple sensitivity metric, called *feature consistency towards transformations (FCT)*, to distinguish poisoned samples from clean samples in the untrustworthy training set. Moreover, we propose two effective backdoor defense methods. Built upon a sample-distinguishment module utilizing the FCT metric, the first method trains a secure model from scratch using a two-stage secure training module. And the second method removes backdoor from a backdoored model with a backdoor removal module which alternatively unlearns the distinguished poisoned samples and relearns the distinguished clean samples. Extensive results on three benchmark datasets demonstrate the superior defense performance against eight types of backdoor attacks, to state-of-the-art backdoor defenses. Codes are available at: https://github.com/SCLBD/Effective_backdoor_defense.

## 1 Introduction

Training deep neural networks (DNNs) often requires a large amount of training data, which is sometimes obtained from a third-party untrustworthy source. However, the untrustworthy data may bring serious security threats. One of the typical threats is the poisoning-based backdoor attack [1], which could inject undesired backdoor—the correlation between trigger(s) and target class(es)—into the model through maliciously poisoning a few training samples. Specifically, as shown in the top left of Fig. 1, each poisoned sample is attached with a trigger (see a small grid patch) at the bottom right corner, and relabelled as a target class. Consequently, the trained backdoored model will predict clean samples very well, but is likely to predict any sample with the trigger to be the target class.

It has been observed in [2] that poisoned samples with triggers are likely to gather together in the feature space of a backdoored model, as shown in the top right of Fig. 1. Note that these poisoned samples contain diverse objects (may be from different source classes), but the information from these objects seems to be ignored by the backdoor model. In other words, the feature representations of poisoned samples are dominated by the triggers, rather than the objects. We conjecture that such a domination is mainly due to the overfitting to the triggers by the backdoor model, since triggers across different poisoned samples are much less diverse than objects. To verify this conjecture, we propose to slightly perturb both poisoned and clean samples, such as

---

∗Corresponding authors.

36th Conference on Neural Information Processing Systems (NeurIPS 2022).

rotation transformation. As shown in the bottom right of Fig. 1, there is no longer gathering of poisoned samples in the feature space, and they are located close to samples of their source classes, *i.e.*, the dominance of triggers over other objects disappears, which verifies the triggers overfitting.

Besides, although the feature representations of clean samples are also affected by transformations, their changes are much smaller than those of poisoned samples. In other words, poisoned samples are more sensitive to transformations than clean samples. It inspires that poisoned samples could be distinguished from clean samples according to the sensitivity to transformations, which is measured by a simple sensitivity metric, called *feature consistency towards transformations (FCT)*. In our experiments, the precision of the distinguished clean and poisoned samples is nearly 100% in most cases, respectively.

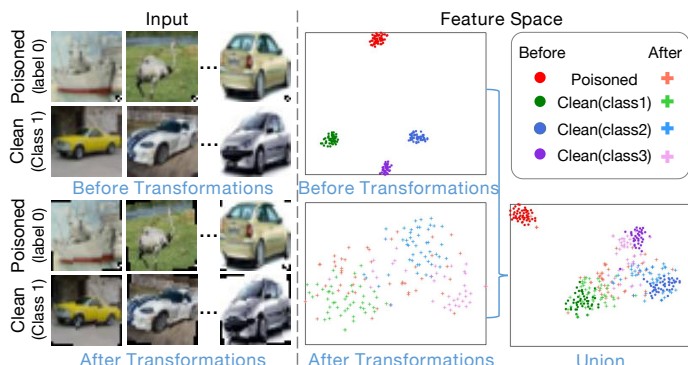

Figure 1: Poisoned and clean samples from CIFAR-10 [3], and the t-SNE [4] visualization of their feature representations with a backdoored model. As shown in the *Union* figure, the changes of poisoned samples (from · to +) are much larger than those of clean samples (from · to + in other colors). Note that we present only three classes for clear illustration.

In this work, we aim to obtain a secure model (*i.e.*, high-performance and without backdoor) based on an untrustworthy training set. To this end, we consider two defense paradigms: one is training a secure model from scratch, while the other is firstly training a backdoored model using standard supervised learning, and then removing backdoor from the backdoored model. Under paradigm 1, we propose an innovative secure training method, called *Distinguishment and Secure Training (D-ST)*, which consists of two consecutive modules. The first *sample-distinguishment (SD) module* splits the whole training set into clean, poisoned and uncertain samples, according to the FCT metric. The second *two-stage secure training (ST) module* firstly learns the feature extractor via semi-supervised contrastive learning, and then learns the classifier via minimizing a mixed cross-entropy loss. Under paradigm 2, we propose an innovative backdoor removal method, called *Distinguishment and Backdoor Removal (D-BR)*, which consists of the SD module and a *backdoor removal (BR) module*. BR module alternatively unlearns the distinguished poisoned samples and learns the distinguished clean samples. Extensive experiments are conducted to verify the superior defense performance of the above two proposed methods, as well as effectiveness of each individual module.

The main contributions of this work are three-folds. **(1)** We demonstrate the sensitivity of poisoned samples to transformations, which is mainly due to the overfitting to trigger, and propose a simple sensitivity metric to distinguish poisoned samples from clean samples. **(2)** We propose two effective backdoor defense methods for training a secure model from scratch and removing backdoor from the backdoored model, respectively. **(3)** Extensive experiments on 3 benchmark datasets show the superior performance of the proposed defense methods against 8 widely used backdoor attacks, to 6 state-of-the-art defense methods.

## 2 Related work

**Backdoor attack.** In poisoning-based backdoor attacks, the attacker attaches a few training samples with *trigger(s)*, and relabel them as *target class(es)*. Existing attacks can be categorized according to a variety of criteria as follows. (1) Size of trigger: *Patch-based attacks* [1, 5, 6] craft patch-like triggers while in *blend-based attacks* [7, 8], triggers capture the whole image. (2) Visibility of trigger: *Visible attacks* [1, 6] design visible but not suspicious triggers while *invisible attacks* [8, 9, 10] propose invisible and still effective ones. (3) Variability of trigger: Triggers are invariant in *sample-agnostic attacks* [1, 7, 11] while vary with samples in *sample-specific attacks* [9, 12]. (4) Label-consistency: If poisoned samples are chosen from samples with target class, then we call these attacks as *clean-label attacks* [11, 13, 12, 14]. Otherwise, we name them as *dirty-label attacks* [1, 6, 7, 8]. (5) Number of

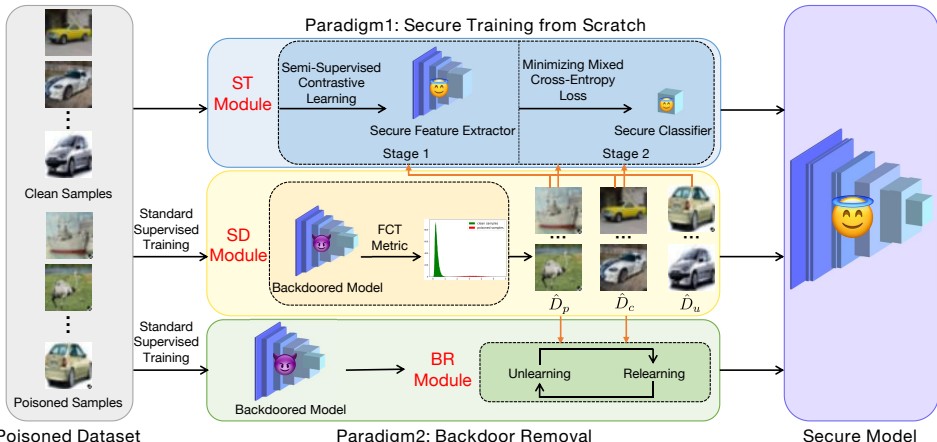

Figure 2: Framework of two proposed backdoor defense methods for secure training from scratch (paradigm1) and backdoor removal (paradigm2), respectively.

target classes: *All2one attacks* [1, 6, 7, 8] designate one class as the target class while in *all2all attacks* [1], poisoned samples are relabelled as the next class. There are also other attacks [15, 16, 17, 18, 19] that require the attacker to control the training process, which are out of scope of this paper. The attack performance of the methods above can be referred to the Backdoorbench [20].

**Backdoor defense.** In general, there are two types of defense paradigms against poisoning-based backdoor attacks—secure training and backdoor removal. Mainstream defense methods belong to the latter one, which leverage the model properties [21, 22, 23] or the feature space characteristics [24, 5, 25, 26, 27, 28, 29] of a backdoored model, to remove the hidden backdoor. For instance, FP [21] observes that some neurons are activated by poisoned samples while others are by clean samples. AC [24] notices the difference in size between the cluster of clean target samples and that of poisoned samples. So far, there is still few works for the former defense. Secure training attempts to train a secure model from scratch, preventing backdoor inserted during training. The core point lies in how to distinguish poisoned samples from clean samples. The first method DBD [2] uses the loss values, the symmetric cross-entropy loss particularly, to distinguish samples during training while our proposed secure training method, *i.e.*, D-ST, leverages the property that the feature representations of poisoned samples in a backdoored model are more sensitive to transformations than those of clean samples, to distinguish samples in advance of the training.

## 3 Proposed method

### 3.1 Problem formulation

**Threat model.** In this paper, we consider the threat model of poisoning-based backdoor attacks, where the attacker can manipulate a few samples in the original clean training set $D_{train} = D_c \cup \{(\boldsymbol{x_i}, y_i)\}_{i=1}^{m_p}$, with $D_c = \{(\boldsymbol{x_i}, y_i)\}_{i=1}^{m_c}$ indicating the unmanipulated subset. For the remaining $m_p$ samples, each sample $\boldsymbol{x}_i \in \mathcal{X}$ is fused with a trigger $\boldsymbol{\delta}$ to form a poisoned sample $\bar{\boldsymbol{x}}_i = \boldsymbol{x}_i \oplus \boldsymbol{\delta}$ with $\oplus$ being the fusion operator. Meanwhile, its label $y_i \in \mathcal{Y}$ is also changed to a target class $t$. Then, a poisoned training set is constructed, denoted as $\bar{D}_{train} \equiv D_c \cup D_p$, with $D_p = \{(\bar{\boldsymbol{x}}_i, t)\}_{i=1}^{m_p}$. When a user downloads $\bar{D}_{train}$ and trains a DNN classifier $g_{\boldsymbol{\theta}} : \mathcal{X} \to \mathcal{Y}$ based on $\bar{D}_{train}$ using the standard supervised learning algorithm, it may learn a undesired backdoor, *i.e.*, a stable mapping from the trigger $\boldsymbol{\delta}$ to the target class $t$. Consequently, for any new sample with the trigger $\boldsymbol{\delta}$, it is likely to be predicted as the target class $t$. Note that the user does not know which sample is poisoned or clean.

**Defense goal.** Given the poisoned training set $\bar{D}_{train}$, the defender aims to obtain a high-performance model $g_{\boldsymbol{\theta}}$ without backdoor, *i.e.*, a secure model. In this work, we consider two different paradigms:

- **Paradigm 1**: a secure model is directly trained from scratch, as described in Section 3.3.
- **Paradigm 2**: a backdoored model is firstly trained using the standard supervised learning, then the backdoor is removed from the backdoored model, as described in Section 3.4.

## 3.2 Sensitivity of poisoned samples

**Sensitivity metric.** As illustrated in Section 1 and Fig. 1, we have found that the poisoned samples are much more sensitive to transformations than the clean samples in a backdoored model. To accurately measure such a difference, we propose a simple metric, *feature consistency towards transformations (FCT)*. Specifically, given a backdoored model $g_{\boldsymbol{\theta}}$ trained on $\bar{D}_{train}$ with $f_{\boldsymbol{\theta}_e}(\cdot)$ indicating its feature extractor, and a set of transformations $\tau$ (*e.g.*, rotation, scaling, will be specified in experiments), for any sample $\boldsymbol{x}$ (poisoned or clean), the FCT metric is formulated as follows:

$$\Delta_{trans}(\boldsymbol{x}; \tau, f_{\boldsymbol{\theta}_e}) = \|f_{\boldsymbol{\theta}_e}(\boldsymbol{x}) - f_{\boldsymbol{\theta}_e}(\tau(\boldsymbol{x}))\|_2^2. \tag{1}$$

It measures the change of the feature representation due to the transformations $\tau$. If $\Delta_{trans}(\boldsymbol{x}; \tau, f)$ is large, then it means that $\boldsymbol{x}$ is sensitive to $\tau$, otherwise stable. For clarity, we use $\Delta_{trans}(\boldsymbol{x})$ hereafter.

**Sample-distinguishment module.** Utilizing FCT, we develop a sample-distinguishment (SD) module. Specifically, we firstly train a backdoored model $g_{\theta}$ based on $\bar{D}_{train}$ using the standard supervised learning algorithm with a few epochs (explained in Appendix A.1). Then, we calculate $\Delta_{trans}(\boldsymbol{x}_i), \forall \boldsymbol{x}_i \in \bar{D}_{train}$, and plot the histogram. As shown in Fig. 3, where two representative backdoor attacks are evaluated, there is remarkable difference on the distribution between the poisoned and the clean samples in both histograms. It demonstrates that $\Delta_{trans}$ is a good metric to distinguish the poisoned samples from the clean samples in $\bar{D}_{train}$. Based

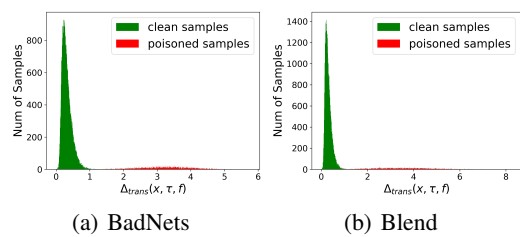

(a) BadNets          (b) Blend

Figure 3: Distribution of clean and poisoned samples with respect to the FCT metric on CIFAR-10.

on the sensitivity histogram, we set two proportion values $\alpha_c, \alpha_p \in [0, 1]$. The samples with the bottom-$\alpha_c$ $\Delta_{trans}$ values are separated to a subset of clean samples $\hat{D}_c$, while those with the top-$\alpha_p$ $\Delta_{trans}$ values are separated into a subset of poisoned samples $\hat{D}_p$, while the remaining samples are partitioned as an uncertain subset denoted as $\hat{D}_u$. We have $\bar{D}_{train} = \hat{D}_c \cup \hat{D}_p \cup \hat{D}_u$. More details are in Algorithm 1 in Appendix A.1

## 3.3 Method for paradigm 1: secure training from scratch

Here, we consider the backdoor defense under paradigm 1. We propose an innovative secure training method, called *Distinguishment and Secure Training* (D-ST) method. As illustrated in Fig. 2, D-ST consists of the SD module (see above) and a **two-stage secure training (ST) module**, which is described as follows. Details of the D-ST method are summarized in Algorithm 3 in Appendix A.2..

**Stage 1: learning feature extractor via semi-supervised contrastive learning (SS-CTL).** Our method is inspired by a recent backdoor defense method called DBD [2], which proposed to learn a good feature extractor $f_{\boldsymbol{\theta}_e}$ based on $\bar{D}_{train}$ using a self-supervised learning algorithm, *i.e.*, contrastive learning (CTL). Consequently, the feature representations of samples with similar appearances will be similar, and poisoned samples with triggers cannot gather together to form the backdoor. Note that all labels have been abandoned in DBD before the extractor learning since there is no way to identify poisoned samples in advance of the learning, leading to the waste of the valuable information contained in clean samples. Fortunately, the proposed SD module could identify some clean samples. Thus, inspired by the supervised contrastive learning (S-CTL) [30], which has shown to learn a feature extractor with better performance than CTL, we propose a novel learning called *semi-supervised contrastive learning (SS-CTL)*, to learn $f_{\boldsymbol{\theta}_e}$ by minimizing the following loss function:

$$\mathcal{L}_{SS-CTL}(\boldsymbol{\theta}_e; \bar{D}_{train}) = \sum_{(\boldsymbol{x}_i, y_i) \in \hat{D}_p \cup \hat{D}_u} \ell_{CTL}\left(f_{\boldsymbol{\theta}_e}(\tilde{\boldsymbol{x}}_i^{(1)}), f_{\boldsymbol{\theta}_e}(\tilde{\boldsymbol{x}}_i^{(2)})\right) \tag{2}$$

$$+ \sum_{\{(\boldsymbol{x}_i, y_i), (\boldsymbol{x}_j, y_j)\} \subset \hat{D}_c} \ell_{S-CTL}\left(f_{\boldsymbol{\theta}_e}(\tilde{\boldsymbol{x}}_i^{(1)}), f_{\boldsymbol{\theta}_e}(\tilde{\boldsymbol{x}}_i^{(2)}), f_{\boldsymbol{\theta}_e}(\tilde{\boldsymbol{x}}_j^{(1)}), f_{\boldsymbol{\theta}_e}(\tilde{\boldsymbol{x}}_j^{(2)}); y_i, y_j\right),$$

where the contrastive loss $\ell_{CTL}$ encourages the two augmented versions $\tilde{\boldsymbol{x}}_i^{(1)}, \tilde{\boldsymbol{x}}_i^{(2)}$ (*e.g.*, cropping, details are introduced in Appendix C) of a sample $\boldsymbol{x}_i$ to be close in the feature space, while the

supervised contrastive loss $\ell_{S-CTL}$ additionally encourages the feature representations of two clean augmented samples from the same class to be close. In this work, we instantiate $\ell_{CTL}$ as the contrastive loss defined in [31] and $\ell_{S-CTL}$ as the SupCon loss defined in [30].

**Stage 2: learning classifier via minimizing the mixed cross-entropy loss.** Given the feature extractor $f_{\boldsymbol{\theta}_e}$ learned in stage 1, we then learn the classifier $h_{\boldsymbol{\theta}_c}$ by minimizing the following mixed cross-entropy (MCE) loss:

$$\mathcal{L}_{MCE}(\boldsymbol{\theta}_c; \hat{D}_c, \hat{D}_p) = \frac{-1}{|\hat{D}_c|} \sum_{(\boldsymbol{x},y) \in \hat{D}_c} \log[h_{\boldsymbol{\theta}_c}(f_{\boldsymbol{\theta}_e}(\boldsymbol{x}))]_y + \frac{\lambda_p}{|\hat{D}_p|} \cdot \sum_{(\boldsymbol{x},y) \in \hat{D}_p} \log[h_{\boldsymbol{\theta}_c}(f_{\boldsymbol{\theta}_e}(\boldsymbol{x}))]_y, \quad (3)$$

where the first term is the standard cross-entropy loss defined based on the distinguished clean samples $\hat{D}_c$, while the second term is the negative cross-entropy loss defined based on the distinguished poisoned samples $\hat{D}_p$, which is used to eliminate the effect of poisoned samples. $\lambda_p \in \mathbb{R}^+$ is a trade-off parameter between two losses.

### 3.4 Method for paradigm 2: backdoor removal

Here we consider the backdoor defense under paradigm 2. We propose an innovative backdoor removal method, called *Distinguishment and Backdoor Removal* (D-BR) method. As illustrated in Fig. 2, D-BR consists of the SD module (see Section 3.2) and a **backdoor removal (BR) module**. The BR module aims to remove the backdoor from the backdoored model, *i.e.*, the backdoor is no longer activated by the trigger, while keeping the high performance on clean samples. To this end, the BR module implements an iterative learning algorithm, which consists of two alternating steps, *i.e.*, *unlearning* and *relearning*. The D-BR method is summarized in Algorithm 4 in Appendix A.3.

**Unlearning.** This step aims to eliminate the effect of the trigger, through unlearning [32] the poisoned samples in $\hat{D}_p$ distinguished by the SD module, as follows:

$$\mathcal{L}_{unlearn}(\boldsymbol{\theta}; \hat{D}_p) = \frac{1}{|\hat{D}_p|} \sum_{(\boldsymbol{x},y) \in \hat{D}_p} \log[g_{\boldsymbol{\theta}}(\boldsymbol{x})]_y. \quad (4)$$

**Relearning.** After conducting the above unlearning step for one epoch, although the effect of poisoned samples is somewhat eliminated, in experiments we find that the performance on clean samples is also degraded to some extent. Thus, we want to relearn the mapping from the clean objects to the ground-truth classes based on clean samples in $\hat{D}_c$ distinguished by the SD module, as follows:

$$\mathcal{L}_{relearn}(\boldsymbol{\theta}; \hat{D}_c) = \frac{1}{|\hat{D}_c|} \sum_{(\boldsymbol{x},y) \in \hat{D}_c} -\log[g_{\boldsymbol{\theta}}(\boldsymbol{x})]_y. \quad (5)$$

Note that both unlearning and relearning are run for one epoch in each round.

## 4 Experiments

### 4.1 Experimental settings

**Attack configurations.** According to the taxonomy described in Section 2, we consider 8 typical poisoning-based backdoor attacks by choosing at least one method from each category, including: BadNets [1] using two attack types (BadNets-all2one, BadNets-all2all), Trojan backdoor attack [6] (Trojan), Blend backdoor attack using two different patterns (Blend-Signal, Blend-Kitty) [2] [7], Clean-label backdoor (CL) [13], Sinusoidal signal backdoor attack (SIG) [11], Sample-specific backdoor attack (SSBA) [9]. We evaluate all attacks on 3 benchmark datasets, CIFAR-10 [3], CIFAR-100 [3] and an ImageNet subset [33, 9], with ResNet-18 [34] as the base model. Poisoning rate is set to 10% in all attacks. Due to the space limit, more implementations details about attacks can be found in Appendix C.3.

**Defense configurations.** We first compare the proposed D-ST method with DBD [2]. Since studies with this secure-training paradigm are limited, we additionally add 2 baselines for comparison which

---

[2]If not specified, 'BadNets/Blend' generally stands for the BadNets-all2one/Blend-signal attack.

are detailed in Section 4.2. We then compare the proposed D-BR method with 5 state-of-the-art methods with the same backdoor-removal paradigm: the standard fine-tuning FT, ANP [22], NAD [35], MCR [36] and ABL [37]. For methods requiring extra clean data, 1% of the clean training samples are provided. Other configurations are set as clarified in the original papers. In summary, we consider 6 state-of-the-art defense methods and 2 additional baselines. More implementations details can be found in Appendix C.4. For our proposed methods, we use $\alpha_c = 20\%$, $\alpha_p = 5\%$ and $\tau =$ rotate+affine in all experiments. Other details can be seen in Appendix C.5.

**Evaluation metrics.** We evaluate the defense performance adopting two commonly used metrics: accuracy on clean samples (ACC) and attack success rate (ASR), *i.e.*, accuracy of predicting poisoned samples as the target label.

## 4.2 Experimental results

**Effectiveness of D-ST method.** We first consider paradigm 1—secure training from scratch. Performance of different defense methods against various attacks on CIFAR-10 and CIFAR-100 is demonstrated in Table 1. An ideal defense method is supposed to increase ACC while keep ASR as low as possible. Thus, a larger ACC-ASR indicates a better method. We mark the best result in boldface. Note that we only report results on successful attacks where ASR is higher than 85%.

Table 1: Comparisons of the D-ST method with 3 secure-training defense methods (%).

| Dataset ↓ | Defense → 
 Attack ↓ | Baseline1 | | Baseline2 | | DBD | | D-ST | |
|---|---|---|---|---|---|---|---|---|---|
| | | ACC | ASR | ACC | ASR | ACC | ASR | ACC | ASR |
| CIFAR-10 | BN-all2one | 83.54 | 2.60 | 91.32 | 99.91 | 92.75 | 100.00 | **92.77** | **0.03** |
| | BN-all2all | 83.95 | 2.72 | 91.59 | 57.39 | 92.95 | 75.21 | **89.22** | **2.05** |
| | Trojan | 83.77 | 5.24 | 93.63 | 99.98 | 92.81 | 100.00 | **93.72** | **0.00** |
| | Blend-Strip | 85.36 | 99.93 | 94.19 | 100.00 | 94.21 | 99.98 | **93.59** | **0.00** |
| | Blend-Kitty | 85.03 | 99.99 | 94.31 | 100.00 | 93.32 | 100.00 | **91.82** | **0.00** |
| | SIG | 85.14 | 99.02 | 94.37 | 99.93 | 94.37 | 99.71 | **90.07** | **0.00** |
| | CL | 85.79 | 10.76 | 94.58 | 98.87 | 94.32 | 99.87 | **90.46** | **6.40** |
| | Avg | 84.65 | 45.75 | 93.43 | 93.73 | 93.53 | 96.40 | **91.66** | **1.21** |
| CIFAR-100 | BN-all2one | 54.48 | 10.41 | 67.62 | 100.00 | 69.08 | 100.00 | **68.43** | **0.12** |
| | Trojan | 56.17 | 12.76 | 71.01 | 100.00 | 72.18 | 99.99 | **68.04** | **0.08** |
| | Blend-Strip | 58.01 | 99.91 | 72.47 | 99.99 | 71.29 | 99.99 | **67.63** | **0.00** |
| | Blend-Kitty | 57.21 | 99.99 | 73.36 | 99.99 | 72.43 | 100.00 | **67.06** | **0.00** |
| | Avg | 56.47 | 55.77 | 71.12 | 100.00 | 71.24 | 99.99 | **67.79** | **0.05** |

DBD fails in most attacks on CIFAR-10 and CIFAR-100, probably due to the failure of the symmetric cross-entropy to distinguish samples. By comparison, the good performance reached by D-ST illustrates the accurate distinguishment from the FCT-based SD module. We additionally introduce two feasible baselines without requiring special knowledge. Baseline1 first uses SimCLR [31] to train the feature extractor and then trains the classifier on the poisoned dataset with standard supervised learning. By comparison, Baseline2 leverages S-CTL [30] to train the feature extractor. We focus on discussing the effect of different extractor-training algorithms on defense performance. More discussions are in the later experiments. Baseline 1 reveals that training extractor without labels may result in low ASR (<5% / <20% in some cases on CIFAR-10 / CIFAR-100), but will sacrifice ACC definitely (84.65% / 56.47% on average). While Baseline 2 demonstrates that training extractor with all labels guarantees high ACC (93.43% / 71.12% on average), but also brings high ASR (93.73% / 100% on average). By contrast, results of D-ST illustrate the effectiveness of ST module in training the feature extractor in a secure way since ACC is high (91.66% / 67.79% on average) and ASR (1.21% / 0.05% on average) is extremely low.

**Effectiveness of D-BR method.** Then, we consider paradigm 2—backdoor removal. Defense performance of different defense methods against various attacks on CIFAR-10 and CIFAR-100 is demonstrated in Table 2. Results on ImageNet is shown in Table 5 in Appendix D.

*Results on CIFAR-10.* We discover that except for FT and MCR, other selected methods generally reduce ACC markedly. Additionally, they have two common disadvantages. (1) They can not take effect on all attacks. For example, ANP can defend against Trojan and Blend attacks (ASR < 1%) while fails in clean-label attacks, i.e. SIG and CL (ASR > 10%). (2) There exists at least one attack that can disable the methods (ASR > 50%). By contrast, the proposed D-BR method overcomes these drawbacks. It not only maintains ACC as large as that of backdoored model, but also reduces ASR to less than 1% on all attacks, verifying the effectiveness of the BR module and the high precision of the distinguishment conducted by the SD module.

Table 2: Comparisons of the D-BR method with 5 backdoor-removal defense methods on CIFAR-10 and CIFAR-100 (%). 'Backdoored' refers to the backdoored model. * denotes methods which require a few (1%) clean training samples.

| Dataset ↓ | Defense → Attack ↓ | Backdoored | | FT* | | ANP* | | NAD* | | MCR* | | ABL | | D-BR | |
|---|---|---|---|---|---|---|---|---|---|---|---|---|---|---|---|
| | | ACC | ASR | ACC | ASR | ACC | ASR | ACC | ASR | ACC | ASR | ACC | ASR | ACC | ASR |
| CIFAR-10 | BN-all2one | 91.64 | 100.00 | 88.99 | 66.79 | 90.03 | 10.54 | 84.46 | 2.13 | 94.21 | 8.29 | 89.36 | 0.19 | **92.83** | **0.40** |
| | BN-all2all | 92.79 | 88.01 | 90.31 | 4.96 | 86.04 | 1.47 | 84.97 | 1.71 | 92.17 | 2.96 | 79.91 | 78.16 | **92.61** | **0.56** |
| | Trojan | 91.91 | 100.00 | 89.86 | 100.00 | 90.89 | 0.81 | 83.29 | 5.04 | 93.90 | 2.58 | 90.18 | 0.23 | **92.21** | **0.76** |
| | Blend-Strip | 92.09 | 99.97 | 89.91 | 93.50 | 88.33 | 0.04 | 83.09 | 13.30 | 91.77 | 17.96 | 58.46 | 0.22 | **92.40** | **0.06** |
| | Blend-Kitty | 92.69 | 99.99 | 90.47 | 99.31 | 84.07 | 0.01 | 84.54 | 28.96 | 94.42 | 7.49 | 79.20 | 2.27 | **92.11** | **0.14** |
| | SIG | 92.88 | 99.69 | 90.81 | 99.87 | 82.43 | 76.32 | 81.00 | 64.72 | 91.82 | 99.04 | 79.94 | 98.84 | **92.73** | **0.24** |
| | CL | 93.20 | 93.34 | 90.03 | 77.44 | 72.57 | 10.90 | 84.46 | 2.66 | 92.13 | 72.01 | 84.39 | 0.31 | **92.08** | **0.00** |
| | Avg | 92.46 | 97.29 | 90.05 | 77.41 | 84.91 | 14.30 | 83.69 | 16.93 | 92.92 | 30.05 | 80.21 | 25.75 | **92.42** | **0.31** |
| CIFAR-100 | BN-all2one | 71.23 | 99.13 | 70.81 | 66.28 | 65.42 | 0.00 | 69.03 | 11.41 | **73.38** | **0.27** | 66.47 | 0.02 | 72.58 | 0.25 |
| | Trojan | 75.75 | 100.00 | 74.21 | 99.94 | 64.52 | 0.03 | 72.11 | 92.21 | 74.51 | 0.12 | 68.12 | 0.00 | **74.52** | **0.00** |
| | Blend-Strip | 75.54 | 99.99 | 73.36 | 99.65 | 67.38 | 0.00 | 71.18 | 95.78 | 73.37 | 0.07 | 49.13 | 0.00 | **74.35** | **0.00** |
| | Blend-Kitty | 75.18 | 99.97 | 72.93 | 99.96 | 69.03 | 0.00 | 71.73 | 99.93 | 73.93 | 20.60 | 47.05 | 0.00 | **72.00** | **0.01** |
| | Avg | 74.43 | 99.77 | 72.83 | 91.46 | 66.59 | 0.01 | 71.01 | 74.83 | 73.80 | 5.27 | 57.69 | 0.01 | **73.36** | **0.07** |

*Results on CIFAR-100.* Although FT and NAD have a relatively high ACC (> 68%), they fail to reduce ASR (91.46% and 74.83% on average). While ANP and ABL can decrease ASR to less than 0.1%, they sacrifice too much ACC (66.59% and 57.69% on average). Among the selected methods, MCR performs the best (ACC = 73.80% on average, ASR < 0.5% in three cases), but it still fails to defend against the Blend-Kitty attack (ASR = 20.60%). Note that MCR requires extra clean data. In contrast, D-BR keeps ACC higher than 72%, while reduces ASR to almost 0% without any extra clean data.

## 4.3 Ablation studies

**Effectiveness of the SD module.** Here, we aim to study the effectiveness of the SD module. Specifically, we will show how our proposed FCT metric, performs better than other metrics, under the backdoor-removal paradigm for illustration. To this end, we select three existing metrics for comparison. Spectral signatures [5] specifies the metric as *the correlation with the top singular vector of the covariance matrix of feature representations*. DBD [2] assigns *symmetric cross-entropy loss* as the metric. The metric used in ABL [37] is *loss value applied with local gradient ascent*. The metric values of clean samples are smaller than those of poisoned samples according to the former two metrics, while larger for the third metric. For fair comparison, we uniformly set $\alpha_c = 20\%, \alpha_p = 5\%$. We first apply the metric-replaced SD module on the poisoned training set, and then conduct the BR module based on the distinguished samples. Results are shown in Fig. 4.

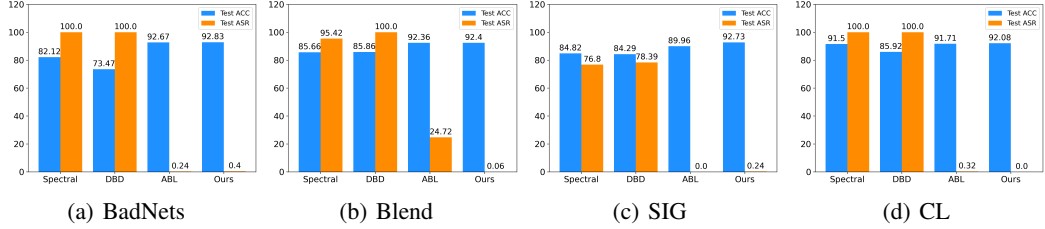

| (a) BadNets | (b) Blend | (c) SIG | (d) CL |

Figure 4: Test ACC and Test ASR of four metric-replaced D-BR methods on the poisoned CIFAR-10.

The height of the blue bar *above* the orange bar suggests how well the metric could distinguish. As shown in Fig. 4 (a,b,d), orange bars are all higher than blue bars for Spectral signatures and DBD, indicating metrics of which fail to distinguish in BadNets, Blend and CL attacks. In contrast, the metric of ABL is reliable since ABL performs well in most cases except for Blend attack where ASR is 24.72%. By comparison, our proposed FCT metric could distinguish samples stably well, resulting in extremely low ASR (< 0.5%) on all attacks. We attribute the success to that FCT exploits the sensitivity of poisoned samples, which is mainly due to the overfitting to trigger by the backdoored model that exists in all backdoor attacks we have evaluated in this paper.

**Effectiveness of the BR module.** Here, we focus on studying the effectiveness of the BR module. Specifically, we aim to show how the iterative learning algorithm consisting of unlearning and relearning performs better than the *pure unlearning* adopted by [37] or *pure relearning*. To this

end, we first conduct the SD module, and then apply different learning algorithms. For the three algorithms, we run 20 epochs on CIFAR-10 and record the variations of Test ACC and Test ASR which are illustrated as Fig. 5.

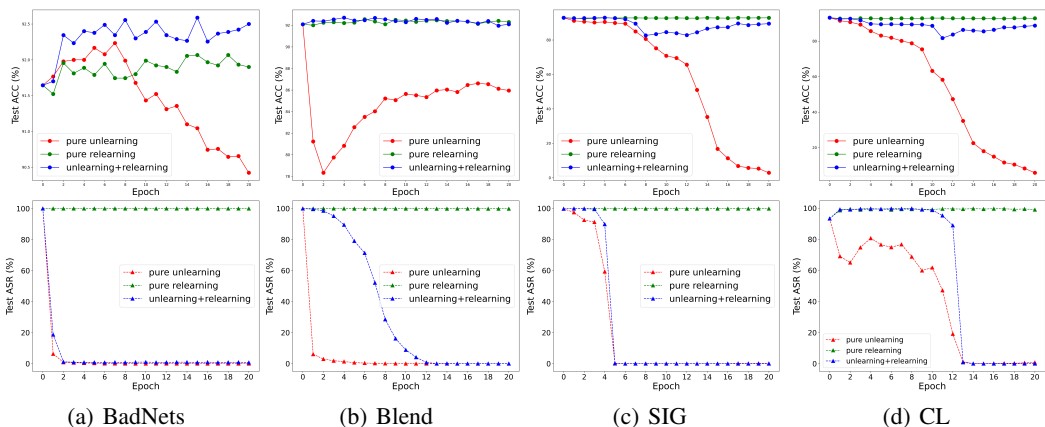

Figure 5: Test ACC(top) and Test ASR(bottom) of three learning algorithms on poisoned CIFAR-10.

Although *pure unlearning* (red lines) effectively decreases ASR, it could hardy maintain ACC, showing a downward trend. The results indicate a strict requirement for choosing the number of unlearning epochs. On the contrary, *pure relearning* (green lines) can keep ACC stably high, but it takes tiny effect in reducing ASR. By contrast, *unlearning+relearning* (blue lines) combines their advantages and successfully diminishes ASR while maintains ACC. ACC and ASR steadily converge to high and low values, respectively, validating the effectiveness and the stability of the BR module.

**Effectiveness of the ST module.** Here, we aim to study the effectiveness of the ST Module. Backdoor can be injected during training the feature extractor $f_{\boldsymbol{\theta}_e}$ and the classifier $h_{\boldsymbol{\theta}_c}$. The final defense performance of $g_{\boldsymbol{\theta}}$ depends on how well $f_{\boldsymbol{\theta}_e}$ and $h_{\boldsymbol{\theta}_c}$ inhibit backdoor.

Firstly, we want to show how SS-CTL performs better than CTL or S-CTL in training a secure feature extractor $f_{\boldsymbol{\theta}_e}$. To this end, we train $f_{\boldsymbol{\theta}_e}$ with different learning algorithms and then uniformly leverage $\mathcal{L}_{MCE}$ to train $h_{\boldsymbol{\theta}_c}$. Training $f_{\boldsymbol{\theta}_e}$ with CTL, as shown in the first row in Table 3, guarantees low ASR, but the low ACC turns into a tradeoff. Note that the low ASR is the joint effort of CTL and $\mathcal{L}_{MCE}$. And the usage of CTL does not indicate low ASR definitely, but it indeed reduces the possibility of backdoor injection in $f_{\boldsymbol{\theta}_e}$ The second row illustrates that S-CTL could bring high ACC and the potentially high ASR, as seen in the SIG and CL attacks. Since all labels (including poisoned) are used in this scenario, ACC is reasonably high. Besides, backdoor is already injected into $f_{\boldsymbol{\theta}_e}$. But due to the inhibition effect of $\mathcal{L}_{MCE}$, the ASR of $g_{\boldsymbol{\theta}}$ may not be high. For example, in dirty-label attacks, i.e. BadNets, Trojan and Blend attacks, ASR is almost 0%. While in clean-label attacks, i.e. SIG and CL attacks, $\mathcal{L}_{MCE}$ can not withstand the backdoor injected in $f_{\boldsymbol{\theta}_e}$, so the ASR is almost 100%. Hence, in order to establish a reliable defense module, $f_{\boldsymbol{\theta}_e}$ should be trained in a more secure way. In comparison, the third row demonstrates the superior defense performance of SS-CTL, illustrating that the ST module securely bridges genuinely clean intra-class samples together which are distinguished by the SD module.

Table 3: Performance with $f_{\boldsymbol{\theta}_e}$ trained with three learning algorithms on the poisoned CIFAR-10.

| Attack → $f_{\boldsymbol{\theta}_e}$ ↓ | BN-all2one | | BN-all2all | | Trojan | | Blend-Signal | | Blend-Kitty | | SIG | | CL | |
|---|---|---|---|---|---|---|---|---|---|---|---|---|---|---|
| | ACC | ASR | ACC | ASR | ACC | ASR | ACC | ASR | ACC | ASR | ACC | ASR | ACC | ASR |
| CTL | 85.63 | 1.52 | 83.02 | 1.65 | 85.03 | 1.32 | 85.12 | 0.00 | 83.49 | 0.00 | 83.10 | 0.00 | 83.77 | 4.88 |
| S-CTL | 92.98 | 0.00 | 93.73 | 0.73 | 93.80 | 0.00 | 94.09 | 0.00 | 94.18 | 0.00 | 94.51 | 99.77 | 94.67 | 98.34 |
| SS-CTL | 92.77 | 0.03 | 89.22 | 2.05 | 93.72 | 0.00 | 93.59 | 0.00 | 91.82 | 0.00 | 90.07 | 0.00 | 90.46 | 6.40 |

Secondly, we explore how $\mathcal{L}_{MCE}$ affects the defense performance of $h_{\boldsymbol{\theta}_c}$. For clarity, we denote $\mathcal{L}_1 \equiv \frac{-1}{|\hat{D}_c|} \sum_{(\boldsymbol{x},y)\in\hat{D}_c} \log[h_{\boldsymbol{\theta}_c}(f_{\boldsymbol{\theta}_e}(\boldsymbol{x}))]_y$ and $\mathcal{L}_2 \equiv \frac{1}{|\hat{D}_p|} \cdot \sum_{(\boldsymbol{x},y)\in\hat{D}_p} \log[h_{\boldsymbol{\theta}_c}(f_{\boldsymbol{\theta}_e}(\boldsymbol{x}))]_y$. We have $\mathcal{L}_{MCE} = \mathcal{L}_1 + \lambda_p \mathcal{L}_2$. Generally, if knowing clean samples, the defender will train $h_{\theta_c}$ with $\mathcal{L}_1$. So here, we aim to show how our proposed $\mathcal{L}_2$ and the trade-off parameter $\lambda_p$ affect $h_{\boldsymbol{\theta}_c}$. To this end, we first fix $f_{\boldsymbol{\theta}_e}$ learned by SS-CTL and then apply $\mathcal{L}_{MCE}$ with $\lambda_p = 0, 0.001, 0.01, 0.1, 1$ on $h_{\boldsymbol{\theta}_c}$. Results are shown in Fig. 6. In the previous experiments, we adopt $\lambda_p = 0.001$.

The comparison between $\lambda_p = 0$ and $\lambda_p \neq 0$ in the right figure illustrates that $\mathcal{L}_2$ can effectively reduce ASR. When comparing different $\lambda_p \neq 0$ in the left figure, we discover that as $\lambda_p$ increases, there is a trend of decrease in ACC. We infer that since $\mathcal{L}_2$ drops faster than $\mathcal{L}_1$, namely unlearning is faster than relearning, adding weights to $\mathcal{L}_2$ makes $h_{\boldsymbol{\theta}_c}$ focus on unlearning instead of relearning, leading to the low ACC. Therefore, we conclude that $\mathcal{L}_2$ helps to inhibit backdoor in

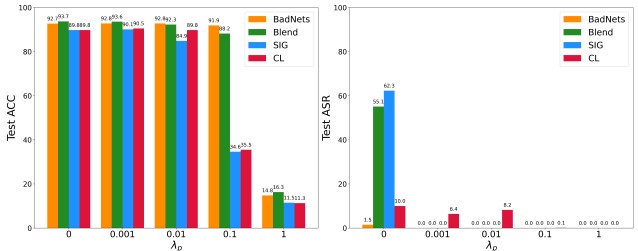

Figure 6: Test ACC (left) and Test ASR (right) under various $\lambda_p$ on the poisoned CIFAR-10.

$h_{\boldsymbol{\theta}_c}$, but its weight should not be too large. $\lambda_p = 0.001$ is considered to be an appropriate choice.

In summary, we have empirically validated the effectiveness of each individual module, and shown the flexibility of our method to combine with other existing modules.

**Appendix.** Due to the space limit, more results and analysis will be presented in Appendix. The overall structure of the Appendix is listed as follows.

- Appendix A: more algorithmic details and analysis on the proposed method.
- Appendix B: more details on semi-supervised contrastive learning (SS-CTL).
- Appendix C: more implementation details.
- Appendix D: results on ImageNet.
- Appendix E: performance with different data transformations $\tau$.
- Appendix F: performance with different proportion values $\alpha_c, \alpha_p$.
- Appendix G: performance with different poisoning rates.
- Appendix H: performance with different model architectures and feature dimensionalities.
- Appendix I: complexities of two proposed methods.

## 5 Conclusions

In this paper, we reveal the sensitivity of poisoned samples to transformations and propose a sensitivity metric, called FCT. Besides, we propose three modules—the SD module to distinguish between clean and poisoned samples, the ST module to train a secure model from scratch and the BR module to remove backdoor—which constitute two defense methods, *i.e.*D-ST and D-BR, to defend under two different defense paradigms. Extensive experiments have demonstrated the effectiveness of each individual module and also the proposed defense methods.

## 6 Broader impact

Poisoning-based backdoor attacks are severe threats to the learning paradigm of learning a DNN model based on the training set from some untrustworthy sources. This work reveals the sensitivity of poisoned samples in the backdoored model, which will help people to better understand the inner mechanism of backdoor attacks. The proposed two effective defense methods can not only significantly mitigate the threat of existing poisoning based backdoor attacks, but also serve as the new baseline for developing more advanced attack methods in future.

## 7 Acknowledgments and disclosure of funding

Baoyuan Wu is supported by the NSFC Fund under grant No.62076213, Shenzhen Science and Technology Program under grant No.RCYX20210609103057050, No.GXWD20201231105722002-20200901175001001, and No.ZDSYS20211021111415025. Haoqian Wang is supported by the NSFC Fund under grant No.61831014, Shenzhen Science and Technology Program under grant No.CJGJZD20200617102601004 and No.JSGG20210802153150005.

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
