# A More algorithmic details and analysis on the proposed method

## A.1 Sample-distinguishment module

We summarize the SD module in Algorithm 1. Note that it is a simplified version for an easy understanding and the detailed version is illustrated in Algorithm 2.

---

**Algorithm 1** A simplified version of sample-distinguishment module

---

**Input:** poisoned training set $\bar{D}_{train}$, a set of data transformations $\tau$, randomly initialized model $g_\theta$, number of training epochs $e_t$, proportion of separated clean (poisoned) samples $\alpha_c$ ($\alpha_p$)

**Output:** separated clean (poisoned) samples $\hat{D}_c$ ($\hat{D}_p$), the remaining uncertain samples $\hat{D}_u$
   # *Inject backdoor into model*
 1: Train a backdoored model $g_\theta$ by minimizing the standard cross-entropy on $\bar{D}_{train}$ for $e_t$ epochs
 2: Let $f(\cdot)$ denote the feature representation of $g_\theta$
   # *Assign threshold for choosing clean and poisoned samples*
 3: Initialize $S \leftarrow \{\}$
 4: **for** each $(\boldsymbol{x}, y) \in \bar{D}_{train}$ **do**
 5:    $S$.append($\Delta_{trans}(\boldsymbol{x}; \tau, f)$)
 6: **end for**
 7: $S' \leftarrow sorted(S)$ in ascending order
 8: $\gamma_c \leftarrow [S']_{\alpha_c|\bar{D}_{train}|}, \gamma_p \leftarrow [S']_{(1-\alpha_p)|\bar{D}_{train}|}$
   # *Separate samples*
 9: **for** each $(\boldsymbol{x}, y) \in \bar{D}_{train}$ **do**
10:    **if** $\Delta_{trans}(\boldsymbol{x}; \tau, f) \leq \gamma_c$ **then**
11:       $\hat{D}_c$.append($(\boldsymbol{x}, y)$)
12:    **else if** $\Delta_{trans}(\boldsymbol{x}; \tau, f) \geq \gamma_p$ **then**
13:       $\hat{D}_p$.append($(\boldsymbol{x}, y)$)
14:    **else**
15:       $\hat{D}_u$.append($(\boldsymbol{x}, y)$)
16:    **end if**
17: **end for**
18: **return** $\hat{D}_c, \hat{D}_p, \hat{D}_u$

---

We omit some algorithmic details and state the SD module in Algorithm 1 for an easy understanding. Here, we continue to elaborate our mechanism in Algorithm 2. The main supplement is the step of fine-tuning the backdoored model, which is highlighted in blue.

Empirically, we discover that $e_t = 2$ is enough for injecting backdoor into model $g_\theta$ for the Train ASR is already higher than 90%. When we then use Algorithm 1 to distinguish samples, we observe that it works under dirty-label attacks (shown in Fig. 8(a,b,d,e)) since the distributions of clean and poisoned samples are quite distinct. However, it doesn't work under clean-label attacks (shown in Fig. 8(c,f)) since poisoned samples are mixed up with clean samples.

We first analyze the reason for the success under dirty-label attacks and then give an assumption of the reason for the failure under the clean-label attacks. In the feature space of a backdoored model [2, 24], poisoned samples are congregated into one cluster, called the *poisoned cluster*, as illustrated in black in Fig. 7. Let $C_c^i, C_p^j$ denote the cluster made up of clean samples labelled as $i$ and poisoned samples labelled as $j$, respectively. We formulate the *intra-cluster distance* between $C_c^i$ and $C_p^j$ as follows.

$$d_{intra}(C_c^i, C_p^j) = \|c_c^i - c_p^j\|_2^2 = \|\frac{\sum_{x_i \in C_c^i} f(x_i)}{|C_c^i|} - \frac{\sum_{x_j \in C_p^j} f(x_j)}{|C_p^j|}\|_2^2, \tag{6}$$

where $c_c^i, c_p^j$ are the center of $C_c^i$ and $C_p^j$, respectively. Under dirty-label attacks, when the trigger is perturbed by the the transformations $\tau$, poisoned samples are no longer located at the poisoned cluster $C_p^t$ (see black points). Instead, they move towards their *true-class clusters* $C_c^i (i \neq t)$ (see non-black points). The long intra-cluster distance between the two clusters, namely $d_{intra}(C_c^i, C_p^t)(i \neq t)$, makes the $\Delta_{trans}(x; \tau, f)$ of poisoned samples large. By contrast, under clean-label attacks where poisoned samples are labeled as their true classes (*i.e.* ground-truth classes), the poisoned cluster $C_p^t$

**Algorithm 2** A detailed version of sample-distinguishment module

---

**Input:** the poisoned training set $\bar{D}_{train}$, a set of data transformations $\tau$, randomly initialized model $g_\theta$, number of training epochs $e_t$, number of fine-tuning epochs $e_{ft}$, proportion of separated clean samples $\alpha_c$, proportion of separated poisoned samples $\alpha_p$, number of classes $N$

**Output:** separated clean samples $\hat{D}_c$, separated poisoned samples $\hat{D}_p$, the remaining uncertain samples $\hat{D}_u$

    *# Inject backdoor into model*
1: Train $g_\theta$ with the standard cross-entropy loss on $\bar{D}_{train}$ for $e_t$ epochs
2: Let $f(\cdot)$ denote the feature representation of $g_\theta$
3: Let $c^i = \frac{\sum_{x_i \in C^i} f(x_i)}{|C^i|}$ denote the center of cluster $C^i$, which consists of samples labelled as $i$
4: Let $\mathcal{L}_{intra} = \frac{1}{N^2} \sum_{i,j \in \{0,1...N-1\}, i \neq j} \frac{<c^i, c^j>}{\|c^i\|_2 \cdot \|c^j\|_2}$
5: Fine-tune $g_\theta$ with $\mathcal{L}_{intra}$ on $\bar{D}_{train}$ for $e_{ft}$ epochs
    *# Assign threshold for choosing clean and poisoned samples*
6: Initialize $S \leftarrow \{\}$
7: **for** each $(\boldsymbol{x}, y) \in \bar{D}_{train}$ **do**
8:     $S$.append($\Delta_{trans}(\boldsymbol{x}; \tau, f)$)
9: **end for**
10: $S' \leftarrow sorted(S)$ in ascending order
11: $\gamma_c \leftarrow [S']_{\alpha_c|\bar{D}_{train}|}, \gamma_p \leftarrow [S']_{(1-\alpha_p)|\bar{D}_{train}|}$
    *# Separate samples*
12: **for** each $(\boldsymbol{x}, y) \in \bar{D}_{train}$ **do**
13:     **if** $\Delta_{trans}(\boldsymbol{x}; \tau, f) \leq \gamma_c$ **then**
14:         $\hat{D}_c$.append($(\boldsymbol{x}, y)$)
15:     **else if** $\Delta_{trans}(\boldsymbol{x}; \tau, f) \geq \gamma_p$ **then**
16:         $\hat{D}_p$.append($(\boldsymbol{x}, y)$)
17:     **else**
18:         $\hat{D}_u$.append($(\boldsymbol{x}, y)$)
19:     **end if**
20: **end for**
21: **return** $\hat{D}_c, \hat{D}_p, \hat{D}_u$

---

(see black points) adjoins $C_c^t$ (see red points) which is also the true-class cluster, leading to the short intra-cluster distance, namely $d_{intra}(C_c^t, C_p^t)$, and the resulting small $\Delta_{trans}(x; \tau, f)$ of poisoned samples.

In order to validate our assumption, we fine-tune the model $g_\theta$ backdoored by clean-label attacks with the *intra-cluster loss* $\mathcal{L}_{intra}^p$ for $e_f = 5$ epochs, which aims to enlarge $d_{intra}(C_c^t, C_p^t)$.

$$\mathcal{L}_{intra}^p = \frac{1}{N} \sum_{i \in \{0,1...N-1\}} \frac{<c_p^t, c_c^i>}{\|c_p^t\|_2 \cdot \|c_c^i\|_2}, \tag{7}$$

where $N$ denote the number of classes. $\mathcal{L}_{intra}^p$ can successfully increase $d_{intra}(C_c^t, C_p^t)$ from 0.25 to 17.19. Then, we reuse the SD module and find that clean and poisoned samples can be well separated.

However, in the realistic scenario, we can not exactly know the poisoned cluster. So, the fine-tuning strategy above is infeasible. Instead, we fine-tune the model $g$ with the *intra-class loss* $\mathcal{L}_{intra}$ for $e_f = 5$ epochs. Denote $C^i$ as the cluster made up of all samples labelled as $i$. $\mathcal{L}_{intra}$ is formulated as:

$$\mathcal{L}_{intra} = \frac{1}{N^2} \sum_{i,j \in \{0,1...N-1\}, i \neq j} \frac{<c^i, c^j>}{\|c^i\|_2 \cdot \|c^j\|_2}, \tag{8}$$

where $c^i = \frac{\sum_{x_i \in C^i} f(x_i)}{|C^i|}$ is the center of $C^i$. Empirically, we discover that $\mathcal{L}_{intra}$ also helps to enlarge $d_{intra}(C_c^t, C_p^t)$, making it increase to 4.96. Subsequently, with the application of the SD module, poisoned samples can be separated from clean samples (Fig. 8(i,l)). Note that $\mathcal{L}_{intra}$ can even boost the separation under dirty-label attacks (Fig. 8(g,h,j,k)). In conclusion, combined with the fine-tuning step, our proposed SD module can work under all kinds of attacks.

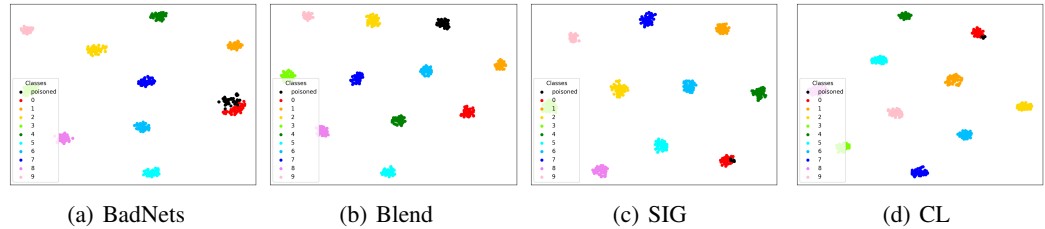

| (a) BadNets | (b) Blend | (c) SIG | (d) CL |

Figure 7: T-SNE [4] visualization of feature space of models backdoored by different attacks on CIFAR-10. Target class is 0. The poisoned cluster is in black while other clean clusters are in non-black.

## A.2 Distinguishment and secure training (D-ST) method

The D-ST method is shown in Algorithm 3.

---

**Algorithm 3** Distinguishment and secure training (D-ST) method

---

**Input:** The training set $\bar{D}_{train}$, number of training epochs for feature extractor $e_e$, number of training epochs for classifier $e_c$

**Output:** A secure model $g_{\boldsymbol{\theta}}$

  *# Sample distinguishment module*
1: Separate $\bar{D}_{train}$ to three subsets, *i.e.*, $\hat{D}_p, \hat{D}_c, \hat{D}_u$ (see Section 3.2)
  *# Secure training module*
2: **Stage 1**: learning the feature extractor $f_{\boldsymbol{\theta}_e}$ through SS-CTL, *i.e.*, minimizing $\mathcal{L}_{SS-CTL}$ on $\bar{D}_{train}$ (see Eq. (2)), for $e_e$ epochs
3: **Stage 2**: learning the classifier $h_{\boldsymbol{\theta}_c}$ via minimizing the mixed cross-entropy loss, *i.e.*, minimizing $\mathcal{L}_{MCE}$ on $\bar{D}_c$ and $\bar{D}_p$ (see Eq. (3)), for $e_c$ epochs
4: **return** $g_{\boldsymbol{\theta}} = h_{\boldsymbol{\theta}_c}(f_{\boldsymbol{\theta}_e}(\cdot))$

---

## A.3 Distinguishment and backdoor removal (D-BR) method

The D-BR method is shown in Algorithm 4.

---

**Algorithm 4** Distinguishment and backdoor removal (D-BR) method

---

**Input:** The training set $\bar{D}_{train}$, number of training epochs $e_t$, number of fine-tuning epochs $e_{ft}$

**Output:** A secure model $g_{\boldsymbol{\theta}}$

  *# Sample distinguishment module*
1: Separate $\bar{D}_{train}$ to three subsets, *i.e.*, $\hat{D}_p, \hat{D}_c, \hat{D}_u$ (see Section 3.2)
  *# Backdoor removal module*
2: Train a backdoored model $g_{\boldsymbol{\theta}}$ by minimizing the standard cross-entropy on $\bar{D}_{train}$ for $e_t$ epochs
3: **for** each $e \in \{1, \ldots, e_{ft}\}$ **do**
4:   **Unlearning**: Fine-tune $g_{\boldsymbol{\theta}}$ by minimizing $\mathcal{L}_{unlearn}$ on $\hat{D}_p$ for an epoch (see Eq. (4))
5:   **Relearning**: Fine-tune $g_{\boldsymbol{\theta}}$ by minimizing $\mathcal{L}_{relearn}$ on $\hat{D}_c$ for an epoch (see Eq. (5))
6: **end for**
7: **return** $g_{\boldsymbol{\theta}}$

---

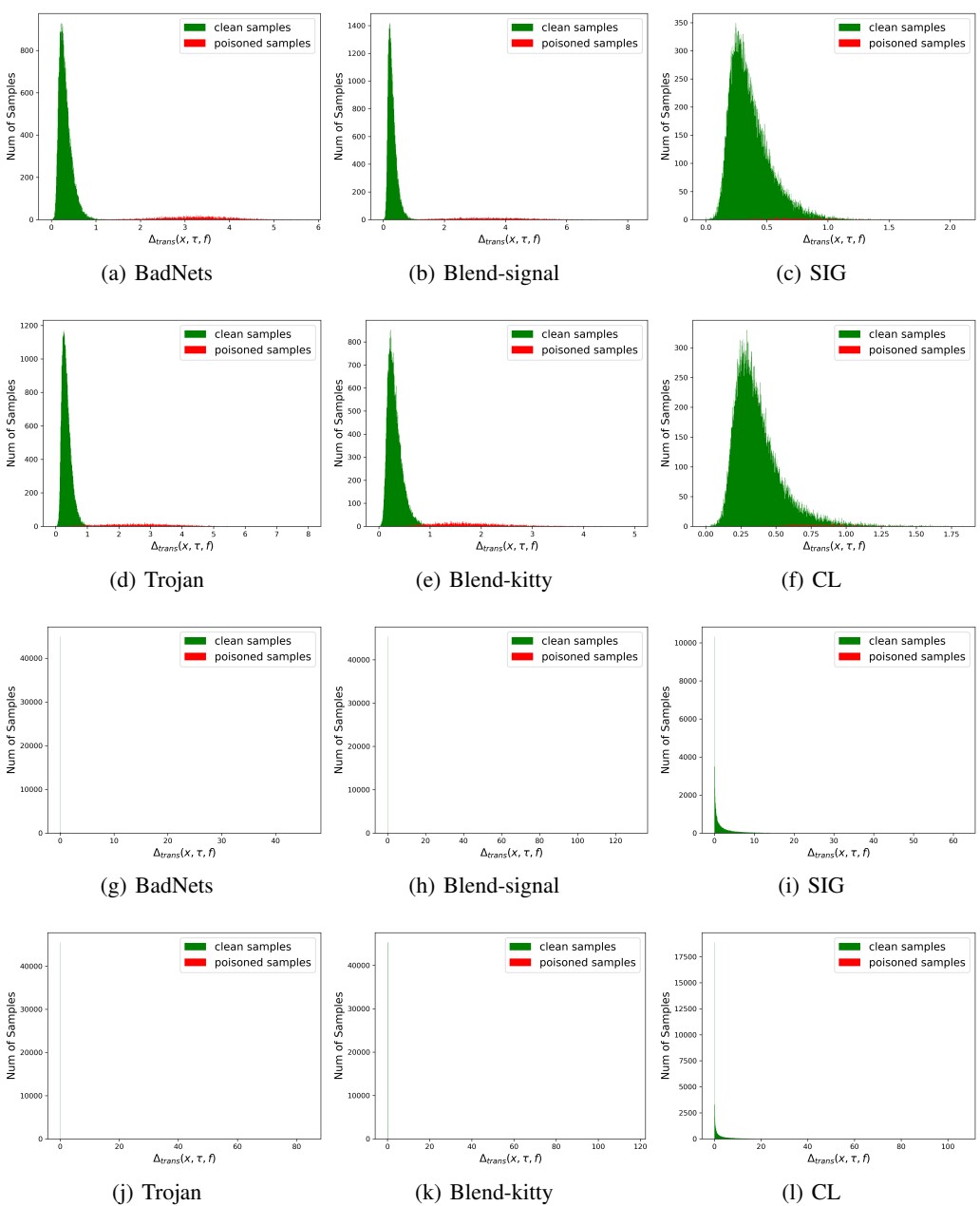

Figure 8: Distribution of clean and poisoned samples with respect to the FCT metric on CIFAR-10. The top(bottom) two rows represent the distribution before(after) the step of fine-tuning. Note that after fine-tuning, clean and poisoned samples are too far apart to be seen clearly.

# B More details on semi-supervised contrastive learning (SS-CTL)

We explain $\ell_{CTL}$ and $\ell_{S-CTL}$ in details here. For clarity, we denote $f_{\boldsymbol{\theta}_e}(\tilde{\boldsymbol{x}}_{\boldsymbol{i}}^{(\mathbf{1})})$ as $f_i^1$ and $f_{\boldsymbol{\theta}_e}(\tilde{\boldsymbol{x}}_{\boldsymbol{i}}^{(\mathbf{2})})$ as $f_i^2$.

$$
\ell_{CTL}\big(f_{\boldsymbol{\theta}_e}(\tilde{\boldsymbol{x}}_{\boldsymbol{i}}^{(\mathbf{1})}), f_{\boldsymbol{\theta}_e}(\tilde{\boldsymbol{x}}_{\boldsymbol{i}}^{(\mathbf{2})})\big) = -\log \frac{\exp\left(f_i^1 \cdot f_i^2 / T\right)}{\sum_{k=1,2}\sum_{j=1}^{|\bar{D}_{train}|} \mathbb{I}(f_i^1 \neq f_j^k)\exp\left(f_i^1 \cdot f_j^k / T\right)}
$$
$$
-\log \frac{\exp\left(f_i^2 \cdot f_i^1 / T\right)}{\sum_{k=1,2}\sum_{j=1}^{|\bar{D}_{train}|} \mathbb{I}(f_i^2 \neq f_j^k)\exp\left(f_i^2 \cdot f_j^k / T\right)}. \tag{9}
$$

If $y_i = y_j$, then

$$
\ell_{S-CTL}\big(f_{\boldsymbol{\theta}_e}(\tilde{\boldsymbol{x}}_{\boldsymbol{i}}^{(\mathbf{1})}), f_{\boldsymbol{\theta}_e}(\tilde{\boldsymbol{x}}_{\boldsymbol{i}}^{(\mathbf{2})}), f_{\boldsymbol{\theta}_e}(\tilde{\boldsymbol{x}}_{\boldsymbol{j}}^{(\mathbf{1})}), f_{\boldsymbol{\theta}_e}(\tilde{\boldsymbol{x}}_{\boldsymbol{j}}^{(\mathbf{2})}); y_i, y_j\big)
$$
$$
= -\frac{1}{3}\{\log \frac{\exp\left(f_i^1 \cdot f_i^2 / T\right)}{\sum_{k=1,2}\sum_{j=1}^{|\bar{D}_{train}|} \mathbb{I}(f_i^1 \neq f_j^k)\exp\left(f_i^1 \cdot f_j^k / T\right)}
$$
$$
+\log \frac{\exp\left(f_i^1 \cdot f_j^1 / T\right)}{\sum_{k=1,2}\sum_{j=1}^{|\bar{D}_{train}|} \mathbb{I}(f_i^1 \neq f_j^k)\exp\left(f_i^1 \cdot f_j^k / T\right)}
$$
$$
+\log \frac{\exp\left(f_i^1 \cdot f_j^2 / T\right)}{\sum_{k=1,2}\sum_{j=1}^{|\bar{D}_{train}|} \mathbb{I}(f_i^1 \neq f_j^k)\exp\left(f_i^1 \cdot f_j^k / T\right)}\}
$$
$$
-\frac{1}{3}\{\log \frac{\exp\left(f_i^2 \cdot f_i^1 / T\right)}{\sum_{k=1,2}\sum_{j=1}^{|\bar{D}_{train}|} \mathbb{I}(f_i^2 \neq f_j^k)\exp\left(f_i^2 \cdot f_j^k / T\right)}
$$
$$
+\log \frac{\exp\left(f_i^2 \cdot f_j^1 / T\right)}{\sum_{k=1,2}\sum_{j=1}^{|\bar{D}_{train}|} \mathbb{I}(f_i^2 \neq f_j^k)\exp\left(f_i^2 \cdot f_j^k / T\right)}
$$
$$
+\log \frac{\exp\left(f_i^2 \cdot f_j^2 / T\right)}{\sum_{k=1,2}\sum_{j=1}^{|\bar{D}_{train}|} \mathbb{I}(f_i^2 \neq f_j^k)\exp\left(f_i^2 \cdot f_j^k / T\right)}\} \tag{10}
$$
$$
-\frac{1}{3}\{\log \frac{\exp\left(f_j^1 \cdot f_i^1 / T\right)}{\sum_{k=1,2}\sum_{j=1}^{|\bar{D}_{train}|} \mathbb{I}(f_j^1 \neq f_j^k)\exp\left(f_j^1 \cdot f_j^k / T\right)}
$$
$$
+\log \frac{\exp\left(f_j^1 \cdot f_i^2 / T\right)}{\sum_{k=1,2}\sum_{j=1}^{|\bar{D}_{train}|} \mathbb{I}(f_j^1 \neq f_j^k)\exp\left(f_j^1 \cdot f_j^k / T\right)}
$$
$$
+\log \frac{\exp\left(f_j^1 \cdot f_j^2 / T\right)}{\sum_{k=1,2}\sum_{j=1}^{|\bar{D}_{train}|} \mathbb{I}(f_j^1 \neq f_j^k)\exp\left(f_j^1 \cdot f_j^k / T\right)}\}
$$
$$
-\frac{1}{3}\{\log \frac{\exp\left(f_j^2 \cdot f_i^1 / T\right)}{\sum_{k=1,2}\sum_{j=1}^{|\bar{D}_{train}|} \mathbb{I}(f_j^2 \neq f_j^k)\exp\left(f_j^2 \cdot f_j^k / T\right)}
$$
$$
+\log \frac{\exp\left(f_j^2 \cdot f_i^2 / T\right)}{\sum_{k=1,2}\sum_{j=1}^{|\bar{D}_{train}|} \mathbb{I}(f_j^2 \neq f_j^k)\exp\left(f_j^2 \cdot f_j^k / T\right)}
$$
$$
+\log \frac{\exp\left(f_j^2 \cdot f_j^1 / T\right)}{\sum_{k=1,2}\sum_{j=1}^{|\bar{D}_{train}|} \mathbb{I}(f_j^2 \neq f_j^k)\exp\left(f_j^2 \cdot f_j^k / T\right)}\}.
$$

Else,

$$
\ell_{S-CTL}\big(f_{\boldsymbol{\theta}_e}(\tilde{\boldsymbol{x}}_{\boldsymbol{i}}^{(\mathbf{1})}), f_{\boldsymbol{\theta}_e}(\tilde{\boldsymbol{x}}_{\boldsymbol{i}}^{(\mathbf{2})}), f_{\boldsymbol{\theta}_e}(\tilde{\boldsymbol{x}}_{\boldsymbol{j}}^{(\mathbf{1})}), f_{\boldsymbol{\theta}_e}(\tilde{\boldsymbol{x}}_{\boldsymbol{j}}^{(\mathbf{2})}); y_i, y_j\big) = 0. \tag{11}
$$

# C More implementation details

All experiments are run on 3 RTX 5000 GPUs and are repeated over 5 runs with different random seeds.

### C.1 Dataset details

We conduct experiments on three benchmark datasets, CIFAR-10 [3], CIFAR-100 [3] and an ImageNet subset [33]. We use the ImageNet subset provided by [9], which contains 200 classes with 100000 training samples (500 samples per class) and 10000 testing samples (50 samples per class).

### C.2 Model details

We conduct experiments with ResNet-18 [34] as base model. For the ST module, we leverage the model provided by [30] on CIFAR-10 and CIFAR-100. For the BR module, we use the model implemented by codes[3] on CIFAR-10, codes[4] on CIFAR-100 and [9] on the ImageNet subset.

### C.3 Attack details

Referring to Section 2, we can categorize existing poisoning-based backdoor attacks according to various criteria. As shown in Table 4, we choose eight attacks to guarantee at least one method in each category. We also give some examples in Fig. 9 to display the patterns of different triggers.

Table 4: Categories of eight backdoor attacks based on different criteria.

| Criterion | Size of trigger | | Visibility of trigger | | Variability of trigger | | Label-consistency | | Num of target classes | |
|---|---|---|---|---|---|---|---|---|---|---|
| | Patch | Blend | Visible | Invisible | Agnostic | Specific | Dirty | Clean | All2one | All2all |
| BadNets-all2one | ✔ | | ✔ | | ✔ | | ✔ | | ✔ | |
| BadNets-all2all | ✔ | | ✔ | | ✔ | | ✔ | | | ✔ |
| Trojan | ✔ | | ✔ | | ✔ | | ✔ | | ✔ | |
| Blend-signal | | ✔ | ✔ | | ✔ | | ✔ | | ✔ | |
| Blend-kitty | | ✔ | ✔ | | ✔ | | ✔ | | ✔ | |
| SIG | | ✔ | ✔ | | ✔ | | | ✔ | ✔ | |
| CL | ✔ | | ✔ | | ✔ | | | ✔ | ✔ | |
| SSBA | | ✔ | | ✔ | | ✔ | ✔ | | ✔ | |

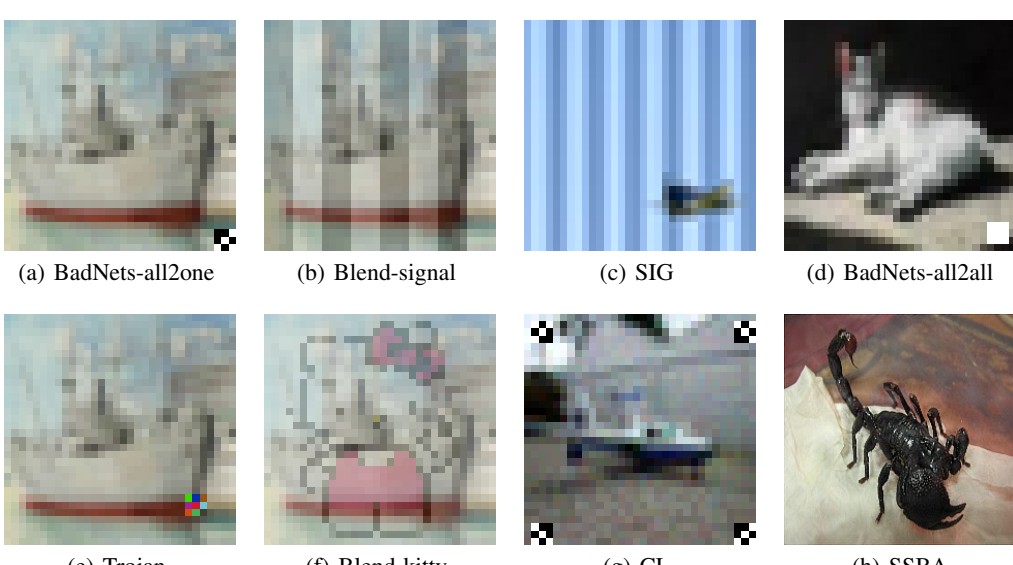

(a) BadNets-all2one  (b) Blend-signal  (c) SIG  (d) BadNets-all2all

(e) Trojan  (f) Blend-kitty  (g) CL  (h) SSBA

Figure 9: Examples of poisoned samples attached with triggers crafted by different backdoor attacks.

Besides, we provide an explanation for the *poisoning rate* configured in attacks. Take CIFAR-10 (50000 training samples) as an example, if the poisoning rate is 10%, then 10% of samples with non-target classes are poisoned in dirty-label (also all2one) attacks (4500 samples) while 10% of samples with target class are poisoned in clean-label (also all2one) attacks (500 samples).

---

[3] https://github.com/huyvnphan/PyTorch_CIFAR10
[4] https://github.com/weiaicunzai/pytorch-cifar100

## C.4 Defense details

Here, we complement some details not mentioned in the main paper.

- FT: We fine-tune the backdoored model on 1% of the clean training samples with 100 epochs.
- MCR: We adopt t = 0.3.
- Baseline1: We first use self-supervised contrastive learning, SimCLR [31] particularly, to train the feature extractor. During the process, data is regarded as unlabeled. Thus, the *positive* of a sample is its augmented version. We then use standard supervised learning to train the classifier on the poisoned dataset.
- Baseline2: We first use supervised contrastive learning, SupContrast [30] particularly, to train the feature extractor. During the process, the *positives* of a sample are its augmented version and samples with the same label (and also their augmented versions). Note that since there are poisoned samples in the dataset, 'samples with the same label' may not be the genuine intra-class samples. We then use standard supervised learning to train the classifier on the poisoned dataset.

## C.5 Details of proposed method

**Choices of parameters.** Here, we complement some details not mentioned in the main paper.

- Algorithm 2: $\tau$ =rotate+affine, $e_t = 2$, $e_f = 10$, $\alpha_c = 0.20$, $\alpha_p = 0.05$. No other data augmentations are used since they would hinder the effect of backdoor implantation.
- Algorithm 3: $e_e = 200$, $e_c = 10$, $\lambda_p = 0.001$. Applied data augmentations are the same as [31].
- Algorithm 4: $e_t = 200$, $e_{ft} = 20$. Data augmentations applied on CIFAR-10 are random crop, random horizontal flip and normalization. Data augmentations applied on CIFAR-100 are random crop, random horizontal flip, random rotation and normalization. Data augmentations applied on ImageNet are random rotation, random horizontal flip and normalization.

**Choices of different data transformations $\tau$.** In Section 4.3, we select six types of $\tau$ to change the pattern or location of the trigger, which are listed as follows and shown in Fig. 10.

- Rotate: Rotate an image by a random degree less than $180°$.
- Affine: Translate an image horizontally and vertically.
- Flip: Flip an image horizontally.
- Crop: Crop an image at a random location.
- Blur: Perform Gaussian blurring on an image by given kernel.
- Erase: Select a random rectangle region in an image and erase its pixels.

# D  Results on ImageNet

Here, we exhibit the performance of our proposed methods on the ImageNet subset [33, 9] in Table 5. Since we aim to study successful backdoor attacks in this paper where ASR of the backdoored model is higher than 85%, results on unsuccessful attacks are not reported. For instance, the increase of number of classes adds to the difficulty of clean-label attacks (ASR < 5%). So, we do not report results on clean-label attacks.

**Performance of the D-ST method.** Note that SupCon [30] does not launch open-souce codes for ImageNet, so the related results are not reported here.

**Performance of the D-BR method.** We do not report results of ANP and MCR on ImageNet since they require particular modification to model and do not have an open-source version for ImageNet. As illustrated in Table 5, among the selected methods, ABL performs the best with ACC as 80.16% and ASR as 23.94% on average. However, it still fails on Blend-Strip attack (ASR = 95.75%). In contrast, D-BR performs steadily well on all attacks, even reducing ASR to 0% in most cases, which demonstrates the effectiveness of the SD module and the BR module.

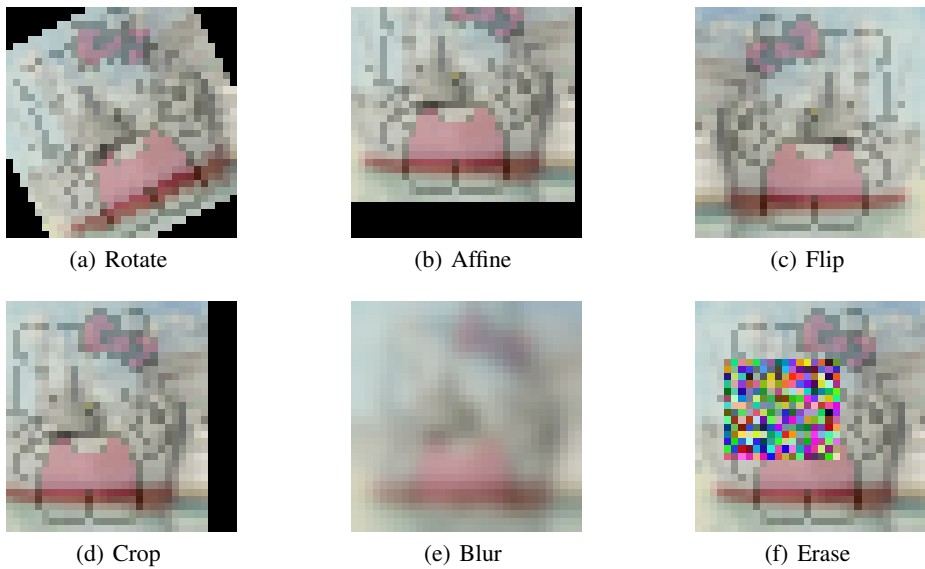

| (a) Rotate | (b) Affine | (c) Flip |
| (d) Crop | (e) Blur | (f) Erase |

Figure 10: Examples of six data transformations.

Table 5: Comparisons of D-BR with 3 backdoor-removal defense methods on ImageNet subset. The best result (with the largest ACC-ASR) is denoted in boldface.

| ImageNet | Backdoored | | FT* | | NAD* | | ABL | | D-BR | |
| Attack | ACC | ASR | ACC | ASR | ACC | ASR | ACC | ASR | ACC | ASR |
|---|---|---|---|---|---|---|---|---|---|---|
| BadNets-all2one | 84.72 | 95.80 | 82.20 | 56.66 | 63.07 | 0.41 | 82.72 | 0.00 | **83.66** | **0.00** |
| Blend-Strip | 84.36 | 97.64 | 82.35 | 78.82 | 59.66 | 16.19 | 79.71 | 95.74 | **80.40** | **0.00** |
| Blend-Kitty | 85.46 | 99.68 | 82.63 | 99.60 | 63.78 | 98.21 | 77.10 | 0.00 | **84.29** | **0.00** |
| SSBA | 85.24 | 99.64 | 82.35 | 97.63 | 63.82 | 34.62 | 81.10 | 0.00 | **83.77** | **0.09** |
| Avg | 84.95 | 98.19 | 82.38 | 83.18 | 62.58 | 37.36 | 80.16 | 23.94 | **83.03** | **0.02** |

# E   Performance with different data transformations

Here, we study how $\tau$ affects the performance of the SD module, specifically the precision of $\hat{D}_c$ (clean-precision) and $\hat{D}_p$ (poison-precision). We select six types of $\tau$ which change the pattern or location of trigger, detailed in Appendix C.5. In order to strengthen the perturbation to trigger, we adopt a combined $\tau$, combining two of the six choices. The total 36 combinations are applied on the poisoned training set and the precision is illustrated in Fig. 11 and Appendix E.

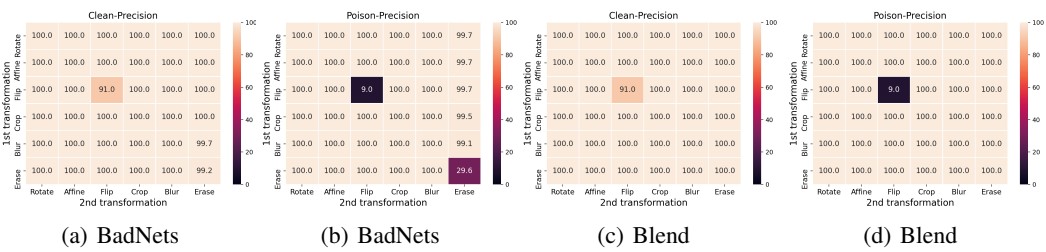

| (a) BadNets | (b) BadNets | (c) Blend | (d) Blend |

Figure 11: Clean(poison)-precision of $\hat{D}_c$ and $\hat{D}_p$ under different $\tau$. Samples are from CIFAR-10.

Apparently, the lighter the color, the higher the precision. One worse case is applying 'flip' twice, equal to not using any transformation. Hence, genuinely poisoned samples can not be identified since trigger is not perturbed, and the poison-precision is extremely low. Another slightly worse case is applying 'erase' as the second transformation. We infer that adding a noise patch on the random position of an image, 'erase' may fail to obscure the trigger. In general, the SD module is

highly effective in identifying genuinely clean and poisoned samples under most combined $\tau$ with 100% precision, also validating the sensitivity of poisoned samples to transformations. Note that the 100% clean-precision does not mean all distinguished clean samples being genuinely clean. In fact, we discover that there are several genuinely poisoned samples, like five, in the distinguished clean samples. These tiny amount of poisoned samples are not reflected on the clean-precision due to the round-off.

Additionally, we show more results in Fig. 12. As a clean-label attack, SIG has high clean-precision but low poison-precision, which is different from BadNets and Blend attacks. We infer that the difference comes from the definition of the attack. Although we set a uniform poisoning rate (10%) for all attacks, the total number of poisoned samples is various. Take CIFAR-10 as an example where there are 50000 training samples (5000 samples per class). In a dirty-label (also all2one) attack, excluding the 5000 samples from target class, there are 4500 poisoned samples totally. By contrast, in a clean-label (also all2one) attack, since the attacker only targets at samples from target class, there are only 500 poisoned samples. Recalling $\alpha_p = 5\%$, we separate 2500 samples as poisoned samples, which are much more than the actual 500 poisoned samples. Therefore, the poison-precision under a clean-label attack is quite low.

Nevertheless, the seemingly adverse fact actually helps to validate one of the benefits of our proposed defense modules (the ST module and the BR module), which is the *robustness to the wrong distinguishment in the SD module*. Specifically, even with a low poison-precision, the proposed two defense methods, *i.e.* D-ST (SD module + ST module) and D-BR (SD module + BR module), are still effective in defending against clean-label attacks, which demonstrates that the two defense modules are robust to the inaccurate distinguishment in the SD module. The robustness probably attributes to the high clean-precision (> 99%). It could help our proposed methods 'survive' under extreme circumstances where there are only tiny amount of poisoned samples, further proved in Appendix G empirically. Hence, despite some newly introduced attacks focusing on reducing the poisoning rate to add to the difficulty in defense, our method is still capable of coping with these potential attacks.

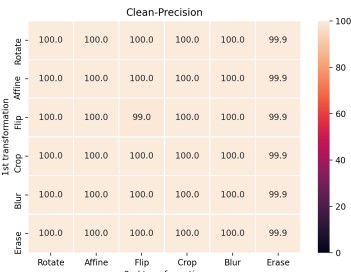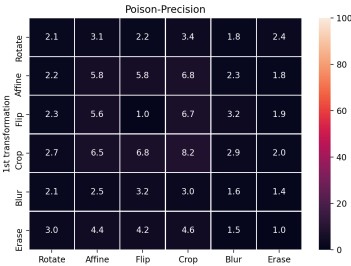

Figure 12: Clean(poison)-precision of the separated clean(poisoned) samples identified by our proposed FCT-based SD module under different $\tau$. Samples are from CIFAR-10 poisoned by SIG attack.

## F    Performance with different proportion values

In this part, we explore how $\alpha_c$ and $\alpha_p$, specified in the SD module, influence the final defense performance. To this end, we first conduct the SD module with different $\alpha_c$ and $\alpha_p$, and then test the performance under the backdoor-removal paradigm for illustration. Recall the settings in the previous experiments where $\alpha_c = 20\%$ and $\alpha_p = 5\%$.

First, we fix $\alpha_p$ as 5% and vary $\alpha_c$ from 0% to 80%. Results are illustrated in Fig. 13 (a). Comparing $\alpha_c = 0$ and others, we see that $\hat{D}_c$ plays an important role in boosting ACC. As shown in the left figure, the model may suffer from extremely low ACC without these samples. As $\alpha_c$ grows, ACC increases steadily and finally converges. However, when $\alpha_c$ is too large (eg. 80%), $D_c$ may contain genuinely poisoned samples, resulting in the rise of ASR, as shown in the right figure. Hence, 20% and 40% are appropriate choices for $\alpha_c$.

Then, we fix $\alpha_c$ as 20% and alter $\alpha_p$ from 0% to 20%. Results are depicted in Fig. 13 (b). The comparisons between $\alpha_p = 0$ and others validate the effect of $\hat{D}_p$ on decreasing ASR. Specifically, as shown in the right figure, ASR could be extremely high if none of these samples are available. While with the increase of $\alpha_p$, ASR declines steadily and finally converges. Nevertheless, excessive $\hat{D}_p$ could hurt the model and lead to a reduction in ACC, as shown in the left figure. We infer that the excessive samples may include a certain amount of genuinely clean samples and unlearning which is deleterious to the model. Therefore, a moderate $\alpha_p$ (eg. 5%) is preferred.

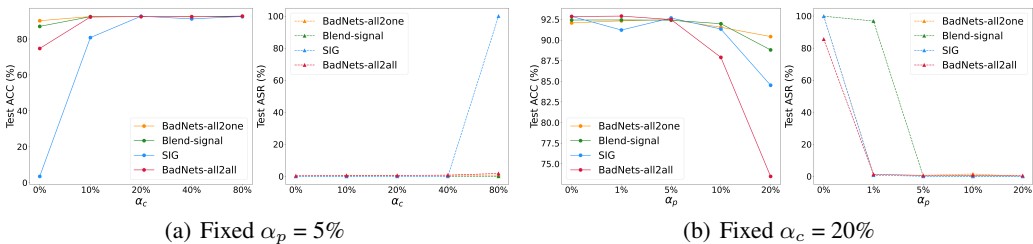

(a) Fixed $\alpha_p = 5\%$        (b) Fixed $\alpha_c = 20\%$

Figure 13: Test ACC and Test ASR of the D-BR method with different $\alpha_p$ and $\alpha_c$ settings in the SD module, on CIFAR-10 poisoned by different attacks. $\alpha_p$ and $\alpha_c$ are fixed separately.

## G    Performance with different poisoning rates

In this section, we aim to show performance of our proposed methods, namely D-ST and D-BR, under different poisoning rates. To this end, we set the poisoning rate as 5% and 20% respectively, and conduct the methods on CIFAR-10 poisoned by different attacks. Here, we choose four representative attacks for illustration. Note that we keep $\alpha_c = 20\%$ and $\alpha_p = 5\%$ as before. The results are shown in Table 6.

Table 6: Performance of our proposed methods on CIFAR-10 poisoned under different poisoning rates.

| Poisoning rate | 5% | | | | | | 20% | | | | | |
|---|---|---|---|---|---|---|---|---|---|---|---|---|
| CIFAR-10 | Backdoored | | D-BR | | D-ST | | Backdoored | | D-BR | | D-ST | |
| Attack | ACC | ASR | ACC | ASR | ACC | ASR | ACC | ASR | ACC | ASR | ACC | ASR |
| BadNets-all2one | 91.31 | 99.98 | 87.52 | 4.22 | 89.38 | 9.44 | 89.06 | 99.99 | 90.32 | 0.96 | 91.48 | 0.13 |
| BadNets-all2all | 91.67 | 83.23 | 91.71 | 0.78 | 89.84 | 1.42 | 91.02 | 87.01 | 91.41 | 0.60 | 89.98 | 1.29 |
| Blend-Strip | 91.70 | 99.99 | 89.07 | 2.49 | 89.56 | 0.00 | 90.78 | 100.00 | 90.11 | 7.42 | 90.72 | 0.00 |
| SIG | 92.00 | 93.26 | 87.9 | 0.75 | 89.56 | 0.63 | 92.00 | 99.87 | 90.76 | 0.44 | 89.13 | 0.19 |
| Avg | 91.67 | 94.11 | 89.05 | 2.06 | 89.59 | 2.87 | 90.71 | 96.72 | 90.65 | 2.36 | 90.33 | 0.40 |

Under the poisoning rate of 20%, D-BR can reduce the average ASR from 96.72% to 2.36%, while keep ACC barely unchanged. Meanwhile, D-ST can train a clean model from scratch with ACC as 90.33% and ASR as 0.40% on average. When the poisoning rate is as small as 5%, D-BR can still manage to decrease ASR by 92.05% on average with a mere drop in ACC (2.62% on average). The model trained with D-ST is relatively clean with ACC as 89.59% and ASR as 2.87% on average. The performance is slightly worse than that of 20%. We infer that in this case, poisoning rate and $\alpha_p$ both equal to 5%. Since we can not find all poisoned samples accurately, the poisoned samples separated by the proposed SD module definitely contain some extent of genuinely clean samples, which is harmful to the performance of the final model. Hence, better performance can be achieved with a careful adjustment in $\alpha_c$ and $\alpha_p$.

## H    Performance with different model architectures and feature dimensionalities

In addition to the ResNet-18 we have tested, here we conduct experiments on another three mainstream model architectures, including ResNet-50, VGG-19[38] and DenseNet-161[39], with the same

parameter setting. Results are shown in Table 7. Besides, we uniformly choose the output of the penultimate layer as the feature representation, resulting in different dimensionalities. Thus, these results can also reflect the sensitivity of our methods to feature dimensionalities.

Table 7: Performance under different model architectures against BadNet attack on CIFAR-10 dataset.

| Model architecture | Dimensionality | Clean-precision of $\hat{D}_c$ | Poison-precision of $\hat{D}_p$ | Backdoored model ACC | ASR | D-BR ACC | ASR | D-ST ACC | ASR |
|---|---|---|---|---|---|---|---|---|---|
| ResNet-18 | 512 | 100% | 100% | 91.64% | 100% | 92.83% | 0.40% | 92.77% | 0.03% |
| ResNet-50 | 2048 | 100% | 9.00% | 90.88% | 100% | 88.47% | 0.00% | 90.32% | 5.89% |
| VGG-19 | 512 | 100% | 100% | 91.09% | 100% | 90.90% | 0.00% | \ | \ |
| DenseNet-161 | 8832 | 99.93% | 21.09% | 90.84% | 100% | 89.82% | 0.00% | \ | \ |

In the following, we will analyze the effectiveness of different modules.

**Effectiveness of SD module**. Both ResNet-18 and VGG-19 achieve 100% precision. In contrast, the poison-precision of ResNet-50 and DenseNet-161 is relatively low, indicating that with the increase in the dimensionality, the gap between the FCT of clean and poisoned samples may be smaller. However, since their clean-precision is as high as about 100% and that our methods are robust to wrong distinguishment as analyzed in Appendix E, this low poison-precision won't significantly influence the final performance which is analyzed subsequently.

**Effectiveness of BR module**. Compared with the performance of the backdoored model, our proposed BR module (*i.e.*, the D-BR method) could reduce ASR from 100% to 0% on the three new architectures. Meanwhile, ACC drops by 2.41% at most.

**Effectiveness of ST module**. Note that since S-CTL [30] (used in stage 1 of D-ST) only released codes for the ResNet architecture, here we do not evaluate the ST module on VGG-19 and DenseNet-161. Compared with the backdoored model which directly employs the supervised learning, our proposed ST module (*i.e.*, the D-ST method) could train a secure model from scratch.

In summary, our methods and the chosen hyper-parameters are generalizable across model architectures. However, we also found that the feature dimensionality may affect the performance, as the Euclidean distance used in our FCT metric may not be suitable for high dimensonal feature space, which will be explored in the future.

# I Complexity of the proposed two methods

Denote the complexity of one forward and backward pass in feature extractor as $a_e$ while that in classifier as $a_c$.

**Complexity of D-ST.** Given the training epochs detailed in Appendix C.5, the complexity is $\mathcal{O}\big(e_e \cdot a_e \cdot [2 \cdot (|\hat{D}_p| + |\hat{D}_u|) + 12 \cdot |\hat{D}_c|^2]\big)$ and $\mathcal{O}\big(e_c \cdot a_c \cdot [|\hat{D}_c| + |\hat{D}_p|]\big)$ for stage 1 and stage 2, respectively. We adopt $e_e = 200$, $e_c = 10$ in the experiments.

**Complexity of D-BR.** The complexity of training a backdoored model is $\mathcal{O}\big(e_t \cdot (a_e + a_c) \cdot |\bar{D}_{train}|\big)$. The complexity of the BR module is $\mathcal{O}\big(e_{ft} \cdot (a_e + a_c) \cdot [|\hat{D}_c| + |\hat{D}_p|]\big)$. We adopt $e_t = 200$, $e_{ft} = 20$ in the experiments.

Take CIFAR-10 as an example. $|\bar{D}_{train}| = 50000$. $|\hat{D}_c|$ and $|\hat{D}_p|$ are approximately 10000 and 2500, respectively. Compared with $a_e$, $a_c$ is relatively small and can be omitted. Thus, the complexity of D-ST is higher than that of D-BR by about $\mathcal{O}\big(12 \cdot a_e \cdot |\hat{D}_c|^2\big)$.