# OpenReview forum: "Effective Backdoor Defense by Exploiting Sensitivity of Poisoned Samples"
_NeurIPS.cc/2022/Conference — NeurIPS 2022 Accept_

### Official Review · Reviewer_tGWV · 2022-06-14

**Rating:** 7
**Confidence:** 4
**Soundness:** 3 good
**Presentation:** 3 good
**Contribution:** 3 good

**Summary:**

The authors observe that the features of poisoned samples are more sensitive to image transformations and use this observation to filter out poisoned data from the training set.  They use this for two different defenses, one to filter out training data before training and one to remove a backdoor from an already backdoored model.  They test against a range of attacks and compare to a variety of defenses.

**Questions:**

N/A

**Limitations:**

The authors refer to Appendix G for limitations which just contains experiments with different percentages of training samples poisoned.  I don't think this fully encompasses limitations of this paper, but I think the limited testing scenarios are clear to a reader, so it's not that big of a deal.

**Strengths And Weaknesses:**

I thank the authors for their interesting work!  This topic is important for security, and I think the NeurIPS community will benefit from new works on defense against poisoning.  The experimental results are pretty strong, and the work compares to other defenses and a range of attacks, so I lean towards accepting.

Feedback can be found below:

For “paradigm 1” did you compare to very strong data augmentations, like mixup/cutmix/maxup, which have been shown to be strong defenses against backdoors or various adversarial training strategies that have been used against backdoor attacks?

It might be good to test against optimization based poisons, such as Sleeper Agent, which don’t have patches on the training samples.  I wonder if these can defeat your defense.  They can also be made adaptive to beat your defense since you could add an objective that makes the features of poisoned samples insensitive to transformations.

Minor point: Full ImageNet does not seem prohibitively expensive here, so it would be good to test there, since it’s a standard setting.

---

> ### Author Response · Authors · 2022-08-01
> **Response to tGWV: Q1**
>
> **Q1:** For “paradigm 1” did you compare to very strong data augmentations, like mixup/cutmix/maxup, which have been shown to be strong defenses against backdoors or various adversarial training strategies that have been used against backdoor attacks?
>
> **A1:** Thanks for your inspiring suggestion in reminding us that the strong data augmentations could be a defense baseline for paradigm 1 (ie. secure training from scratch), which have been shown effective in defending against various attacks. To this end, we apply different strong augmentations on the poisoned dataset and train a model on the augmented data with the standard supervised learning to see whether the augmentations could prevent backdoor inserted into the model. Specifically, we choose three types of strong augmentations—Mixup, Cutout and CutMix, following implementation in https://github.com/facebookresearch/mixup-cifar10/tree/main, https://github.com/uoguelph-mlrg/Cutout, and https://github.com/clovaai/CutMix-PyTorch, respectively. Since data augmentations manipulate data in the input space, the properties (eg. pattern, distribution) of the trigger may have a large influence on the effectiveness of the augmentations. Hence, here we select three types of poisoning-based backdoor attacks—the patch-based attack BadNets (where the trigger pattern is a patch), the blend-based attack Blend (where the trigger pattern is an image) and the clean-label attack SIG (where the trigger is only attached on samples of target class). Results on the CIFAR-10 dataset are shown in Table 11.
>
> **Table 11: Comparison of D-ST and three strong data augmentations (Mixup, Cutout and CutMix) against three attacks on CIFAR-10.** **The best result (with the largest ACC-ASR) is denoted in bold.**
>
> | Attack  | Backdoored model (ACC) | Backdoored model (ASR) | D-ST (ACC) | D-ST (ASR) | Mixup (ACC) | Mixup (ASR) | Cutout (ACC) | Cutout (ASR) | CutMix (ACC) | CutMix (ASR) |
> | ------- | ---------------------- | ---------------------- | ---------- | ---------- | ----------- | ----------- | ------------ | ------------ | ------------ | ------------ |
> | BadNets | 91.64%                 | 100%                   | **92.77%** | **0.03%**  | 91.59%      | 100%        | 92.46%       | 100%         | 92.02%       | 100%         |
> | Blend   | 92.69%                 | 99.99%                 | **91.82%** | **0.00%**  | 92.09%      | 100%        | 92.71%       | 100%         | 92.86%       | 99.98%       |
> | SIG     | 92.88%                 | 99.69%                 | **90.07**  | **0.00%**  | 92.14%      | 99.90%      | 92.78%       | 99.68%       | 93.12%       | 99.81%       |
> | Avg     | 92.40%                 | 99.89%                 | **91.55%** | **0.01%**  | 91.94%      | 99.97%      | 92.65%       | 99.89%       | 92.67%       | 99.93%       |
>
> **Table 12: Comparison of ASR** **in the first training epoch with different augmentation** **against three attacks on CIFAR-10.**
>
> | Augmentation \ Attack | BadNets | Blend  | SIG    | Avg    |
> | --------------------- | ------- | ------ | ------ | ------ |
> | None                  | 100%    | 99.62% | 99.21% | 99.61% |
> | Simple                | 97.81%  | 99.21% | 87.33% | 94.78% |
> | Mixup                 | 47.38%  | 98.83% | 26.96% | 57.72% |
> | Cutout                | 18.90%  | 96.78% | 20.72% | 45.47% |
> | CutMix                | 1.48%   | 93.47% | 22.17% | 39.04% |
>
> **Backdoored model:** If we directly apply the standard supervised learning on the poisoned dataset, we would obtain a backdoored model. As shown in Column 2,3 in Table 11, the average ACC is 92.40% while the average ASR is 99.89%.
>
> **D-ST:** By contrast, if we use our proposed D-ST to train a model from scratch, we could obtain a secure model. As shown in Column 4,5 in Table 11, the average ACC is 91.55% while the average ASR is 0.01%, demonstrating that D-ST could effectively prevent backdoor from inserting into the model while maintaining the high ACC.
>
> **Strong data augmentations:** However, when applying the three strong data augmentations, we discover that the performance of the trained models are similar to that of the backdoored model, all with high ACC and ASR as shown in Column 6-11 in Table 11. In this sense, it seems that these augmentations could not inhibit backdoor. Although the final ASR is high, we discover that the ASR in the first training epoch of strong data augmentations is lower than that of simple augmentations (eg. cropping, flipping) or without augmentation, as shown in Table 12. It tells that although failed to inhibit backdoor, the data augmentations could slow down the backdoor effect, since they increase the diversity of the trigger, which makes the backdoor harder to learn.

---

> ### Author Response · Authors · 2022-08-01
> **Response to tGWV: Q2**
>
> **Q2:** It might be good to test against optimization based poisons, such as Sleeper Agent, which don’t have patches on the training samples. I wonder if these can defeat your defense. They can also be made adaptive to beat your defense since you could add an objective that makes the features of poisoned samples insensitive to transformations.
>
> **A2:** Thanks for your insightful suggestion. We notice that the goal of the Sleeper Agent is to make samples from one particular source class misclassified as a target class when attached with the trigger, which is different from the goal of the mainstream attacks. Hence, we calculate ASR only on triggered source testing samples, instead of the entire testing set. Since Sleeper Agent is an optimization-based method, we assume the attacker is in a white-box setting, where he knows the model architecture used by the defender. With the data augmentations provided in the open source code, we could train a backdoored model with ACC=90.99% and ASR=71.20%. We then conduct our proposed D-BR to see whether it could remove the backdoor. The defense performance is shown in Table 13. Since there is no explicit or implicit objective to make features of poisoned samples insensitive to transformations, our defense could still be effective against such attack, where no patches are on the training samples.
>
> **Actually in our paper, we have tried a similar attack (SSBA [9]) to see how our defense methods perform against attacks where there are no patches on the poisoned training samples**. An example poisoned by SSBA is shown in Figure 9 (h) in Appendix C.3, which seems not different from the natural image to the human eye, similar to that by Sleeper Agent. Since SSBA takes effect on large-scale images such as ImageNet, we test our defense methods against SSBA attack on the ImageNet. As shown in Table 4 in Appendix D, our proposed method could effectively reduces ASR from 99.64% to **0.09%** while keeping the ACC as high as **83.77%**. For more analysis, please refer to Appendix D.
>
> In fact, in order to illustrate that our proposed methods are not designed for any particular attack, we apply them to defend against various types of attacks. Particularly, we categorize existing poisoning-based backdoor attacks according to 5 criteria (*ie. size* *of trigger, visibility of trigger, variability of trigger, label-consistency, number of target classes*) as mentioned in Section 2, and choose 8 typical attack methods in the experiments which cover all categories under every criterion. The type of attack you have mentioned has also been considered in this paper. More details can be seen in Section 2 and Appendix C.3.
>
> **Table 13: Performance of backdoored model and D-BR against attacks with trigger invisible to human eye.**
>
> | Dataset  | Attack        | Backdoored model (ACC) | Backdoored model (ASR) | D-BR (ACC) | D-BR (ASR) |
> | -------- | ------------- | ---------------------- | ---------------------- | ---------- | ---------- |
> | CIFAR-10 | Sleeper Agent | 90.99%                 | 71.20%                 | 89.92%     | 6.90%      |
> | ImageNet | SSBA          | 85.24%                 | 99.64%                 | 83.77%     | 0.09%      |
>
> Last but not least, we are so grateful for your suggestion that we may consider an attack which is be made adaptive to beat out defense since he could add an objective that makes the feature representations of poisoned samples insensitive to transformations. To this end, we consider **an adaptive attack** to see whether our proposed method is robust to such attack. As for the setting of the adaptive attack and the defense performance of our methods against it, please refer to **Response to FGt5: Q3**.
>
> Besides, in our paper, we conducted experiments on an ImageNet subset, which are detailed in Appendix D. And due to the limited rebuttal period, we will add the experiments on the full ImageNet into the revised manuscript. Thanks for your suggestions once again.

---

> > ### Comment · Reviewer_tGWV · 2022-08-06
> > **Rebuttal Acknowledged**
> >
> > Thanks for your detailed response to my feedback.  I have increased by score to a 7.

---

> > > ### Author Response · Authors · 2022-08-06
> > > **Thanks for your encouraging comments**
> > >
> > > Dear Reviewer,
> > >
> > > Thanks for your encouraging and constructive comments, which are really helpful to improve our work.

---

### Official Review · Reviewer_gz1a · 2022-07-10

**Rating:** 5
**Confidence:** 4
**Soundness:** 2 fair
**Presentation:** 3 good
**Contribution:** 2 fair

**Summary:**

Through the observation that clean and backdoored data with a dissimilar feature representation after data transformation techniques (e.g. rotation, scaling), the author proposed a sensitivity metric, feature consistency towards transformations (FCT), to detect the potential backdoor samples. The author further proposed two backdoor removing modules with inspiration from the existing defenses of semi-supervised learning and backdoor unlearning. Extensive results show that the proposed methods outperform the existing backdoor defenses either in backdoor detection or backdoor removal.


**Questions:**

At the same time, I have some concerns about the paper's contributions:

- At a first glance, the proposed methods seem as a unified framework including backdoor detection and backdoor defense forming an integral defense pipeline from training to post-training phase. But after a depth-reading, I consider this work is much like a patchwork combining existing techniques. For instance, the proposed feature detection matrix has been reflected in [1], and the other two defense modules ST and BR are much related to the DBD [2] and ABL [3] respectively. So, please explain clearly the main contribution and what specific contribution\novelty of this work compared with others.

- In addition, I doubt the generality of this method, for which the underlying assumption is that there are differences in feature representation between backdoor samples and clean samples. In fact, this assumption is often related to the design of backdoor models, and trigger types (static trigger, dynamic trigger, optimization-based trigger, etc). For more powerful adaptive attacks such as FC [4], dynamic [5], and latent modification [6], this assumption is no longer valid. In this case, how to ensure adequate defense? The author is invited to give an explanation and further experimental evidence.

[1] Li Y, et al. Rethinking the trigger of backdoor attack. arXiv preprint arXiv:2004.04692, 2020.
[2] Huang K, et al. Backdoor defense via decoupling the training process. ICLR, 2022.
[3] Li Y, et al. Anti-backdoor learning: Training clean models on poisoned data. NeurIPS, 2021.
[4] Shafahi A, Huang W R, Najibi M, et al. Poison frogs! targeted clean-label poisoning attacks on neural networks. NeurIPS, 2018.
[5] Nguyen T A, Tran A. Input-aware dynamic backdoor attack. NeurIPS 2021.
[6] Doan K, Lao Y, Li P. Backdoor attack with imperceptible input and latent modification. NeurIPS, 2021.

**Limitations:**

It is still unclear that the underlining assumption behind ‘feature consistency' is supported efficiently by the proposed solutions with only empirical results. It needs some explanation about this result to backup this drawback.


**Strengths And Weaknesses:**

This paper is well-organized and the main insight is the observation of feature separation by data transformations. The defense solution including a secure training (ST) module and a backdoor removal (BR) module seems simple yet effective against various classical backdoor attacks.

---

> ### Author Response · Authors · 2022-08-01
> **Response to gz1a: First part of Q1**
>
> **Q1:** Please explain clearly the main contribution and what specific novelty of this work compared with others.
>
> **A1:** We appreciate the comment that "this work is much like a patchwork combining existing techniques". It reminds us to clarify the key differences with existing works more clearly, such that the novelties of our method can be highlighted. Here, we would like to explain the key differences with the mentioned 3 existing works individually.
>
> - *Differences between "[1] Rethinking the trigger of backdoor attack. Arxiv 2020." (Rethinking) and our proposed feature consistency towards transformations (FCT)*. **(1) Different observation perspectives**: Rethinking locally changes the trigger to observe the change of the attack success rate (ASR), i.e., observing "the property of existing attacks with static trigger"; our FCT globally transforms both poisoned and clean images to observe their changes of the feature representations in the backdoored model, to further observe the different sensitivities to transformation between poisoned and clean images, i.e., observing the difference between poisoned and clean images. **(2)  Different observation conclusions**: Rethinking concludes that "the backdoor attack is sensitive to the difference between the training trigger and the testing trigger" (the original sentence in [1]); our FCT concludes that "the poisoned samples are much more sensitive to transformations than the clean samples in a backdoored model" (the original sentence in our manuscript), and our FCT metric provides a quantitative measure of this sensitivity. **(3) Different usages**: Rethinking utilizes the trigger sensitivity in two ways, including "the transformation-based defense is defined as introducing a transformation-based pre-processing module on the testing image before prediction" and augmenting the poisoned samples with transformations to enhance its transformation robustness. Our FCT metric is used to distinguish poisoned samples from clean samples, which plays the key role in the following backdoor defense methods (i.e., D-ST and D-BR). We believe above comparisons are sufficient to demonstrate the instrinsic difference between [1] Rethinking and our FCT metric.
>
> - *Differences between "[2] Backdoor defense via decoupling the training process. ICLR, 2022." (DBD) and our Distinguishment and Secure Training (D-ST) method*: Both DBD and D-ST aim to inhibit backdoor during training (ie. realize secure training). However, there are several significant differences. **(1) Different principles of identifying poisoned samples:** **The core point** of secure training is distinguishing poisoned samples from clean samples in the training dataset. The identification principle of DBD is that the loss values of poisoned samples are larger those that of clean samples when training the classifier (the second stage of DBD).  Our identification principle is that the feature representations of poisoned samples in a backdoored model are more sensitive to transformations than those of clean samples, which is measured by our proposed FCT metric. It is clear that **our D-ST is different with DBD on this core point of a secure training-based backdoor defense**. Besides, as evaluated in Section 4.3 (Line 48-265 and Figure 4), our FCT principle can identify different types of poisoned samples more stably than the DBD's principle.  **(2) Different usages and roles of the identified poisoned samples.** In DBD, since the poisoned samples are identified at the end of the second stage (i.e., after training the classifier), they can only be used for semi-supervised fine-tuning via removing the labels of the identified samples to improve the clean accuracy at the last stage. In our D-ST, since the poisoned samples are identified at the beginning, we can utilize them in all training stages. **First**, they are utilized to train a good feature extractor via semi-supervised contrastive learning (SS-CTL), based on labels of the idenfied clean samples. As evaluated in Section 4.3 (Line 281-195 and Table 3), the defense performance using SS-CTL is much better than that of using CTL (which is utilized by DBD), especially on clean accuracy. **Second**, the identified poisoned and clean samples are utilized to design the mixed cross-entropy loss (MCE loss, see Line 167 and Eq. (3)) to learn the classifier. As evaluated in Section 4.3 (Line 296-315 and Figure 6), both loss functions in MCE are important for the overall defense performance. It is clear that the poison identification plays the secondary role in DBD (learning the backbone using CTL plays the key role), while the poison identification plays the key role in our D-ST method. **In summary, there are intrinsic differences on the design logic, the key technology, between DBD and D-ST.** These differences are further reflected by the different defense performance. As shown in Section 4.2 and Table 1, D-ST performs much better than DBD.

---

> > ### Comment · Reviewer_gz1a · 2022-08-06
> > **My concerns have been partially addressed.**
> >
> > I thank the authors for their response, which partially addressed my concerns. In particular, I appreciate the efforts of the authors that have clarified the specific novelty of this work. As such, I have raised my score accordingly.

---

> > > ### Author Response · Authors · 2022-08-06
> > > **Thanks for your encouraging comments**
> > >
> > > Dear Reviewer,
> > >
> > > Thanks for your encouraging and constructive comments, and they are very helpful for us to improve this work.
> > >
> > > Sincerely,
> > > Authors

---

> ### Author Response · Authors · 2022-08-01
> **Response to gz1a: Second part of Q1**
>
> - *Differences between "[3] Anti-backdoor learning: Training clean models on poisoned data. NeurIPS, 2021." (ABL) and our Distinguishment and Backdoor Removal (D-BR) method*: ABL and D-BR have the similar defense strategy that firstly learning the backdoor, then identifying the poisoned samples according to some principles, and finally removing the backdoor based on these samples. ABL and D-BR are different on the key step, i.e., **different** **principles of identifying poisoned samples**. Specifically, ABL utilizes the observation that the model fits poisoned samples much faster than clean samples, i.e., the former loss values decrease faster. Then, ABL designed a local gradient ascent technique to identify the poisoned samples according to the loss values at the early training epochs. In contrast, we utilize the observation that the feature representations of poisoned samples in a backdoored model are more sensitive to transformations than those of clean samples, which is measured by our proposed FCT metric. As evaluated in Section 4.3 (Line 48-265 and Figure 4), our FCT principle can identify different types of poisoned samples more stably than the ABL's principle. Moreover, the iterative relearning and unlearning mechanism in our D-BR method is also very important for the overall defense performance. As evaluated in Section 4.3 (Line 266-271 and Figure 5), it performs much better than the pure relearning and pure unlearning mechanism.
>
> **In summary**, although our methods and existing methods have the same goals (i.e., identifying poisoned samples, inhibiting the formation of backdoor or removing backdoor),  their key techniques and the overall designs to achieve these goals are intrinsically different. We believe that above comparisons have clearly demonstrated these differences, which will be added into the revised appendix, to highlight the novelty and contribution of our methods to the backdoor learning community. We appreciate the reminding from the reviewer once again.

---

> ### Author Response · Authors · 2022-08-01
> **Response to gz1a: First part of Q2**
>
> **Q2:** For attacks such as FC, dynamic, and latent modification, are the proposed methods still effective?
>
> **R2:** Thanks for your constructive suggestion.
>
> **Clarification of threat model:** We would like to firstly clarify that, as clearly described in Line 102 in Section 3.1, "*we consider the threat model of poisoning-based backdoor attacks*", where the attacker can only manipulate the training data. In contrast, some suggested works require control over the training process, *i.e.*, the threat model of *training-controllable backdoor attacks*, which are out of scope of our work.
>
> **Descriptions of three suggested works:** We firstly describe the general information of each suggested attack, as follows:
>
> - *FC: Poison Frog! Targeted Clean-Label Poisoning Attacks on Neural Networks, NeurIPS 2018.* As clearly claimed in the Abstract of its orginal paper, *"they control the behavior of the classifier on a specific test instance without degrading overall classifier performance"*, it is not backdoor attack. Thus, it shouldn't be used to evaluate our backdoor defense methods.
>
> - *Dynamic: Input-aware Dynamic Backdoor Attack, NeurIPS 2020.* This work aims to jointly train a generator which generates input-aware triggers, and a backdoored model which can be activated by the generated triggers during testing. It belongs to the *training-controllable backdoor attack*, as we mentioned above. We find that the code of the github repository https://github.com/VinAIResearch/input-aware-backdoor-attack-release has been removed by its authors. Thus, in the later experiments, we adopt the implementation provided by the BackdoorBench (https://github.com/SCLBD/backdoorbench).
>
> - *Latent modification: Backdoor Attack with Imperceptible Input and Latent Modification, , NeurIPS 2021.* This work also assumes the attacker could have access to the model used by the defender, including both structures and parameters, and could insert the backdoor into model during training. It also belongs to the *training-controllable backdoor attack*. However, the code of this work has not been released. We checked the first author's homepage, there was a code link under this work, but it was linked with a repository of another ICCV 2021 work "LIRA: Learnable, Imperceptible and Robust Backdoor Attacks" (https://github.com/khoadoan106/backdoor_attacks). Due to the limited rebuttal period, we cannot guarantee to correctly implement this method correctly from the scratch, only according to the original paper. Hope the reviewer and Area Chair could understand this difficulty.

---

> ### Author Response · Authors · 2022-08-01
> **Response to gz1a: Second part of Q2**
>
> **Defense against the input-aware backdoor attack:**
>
> **(1) Experimental setting:**
> Due to the different threat model, we have to firstly adjust the experiment, as follows:
>
> - *Step 1: generating the poisoned training dataset*: we conduct the input-aware attack on the same training dataset (*i.e.*, CIFAR-10) and the same model architecture (*i.e.*, ResNet-18) with the subsequent defense step. The attack could output both a backdoored model and a trigger generator. We generate triggers for 10% of the training samples, which form the poisoned training dataset.
>
> - *Step 2: baseline attack:* we train a backdoored model based on the above dataset using the standard supervised learning with 200 epochs. It serves as the baseline to measure the performance of our method.
>
> - *Step 3: our defense:* we conduct our two defense methods (D-BR and D-ST) to obtain secure models based on the above dataset.
>
> **Table 10: Performance of the backdoored model and our proposed methods against the input-aware attack on CIFAR-10.**
>
> |      | Backdoored model | D-BR   | D-ST   |
> | ---- | ---------------- | ------ | ------ |
> | ACC  | 92.23%           | 89.69% | 89.08% |
> | ASR  | 75.96%           | 2.34%  | 9.03%  |
>
> **(2) Experimental results and analysis:**
>
> - *Baseline backdoored model*: As shown in Column 2 of Table 10, the backdoored model gives 92.23% ACC and only 75.96% ASR, indicating the attack is not very strong. Note that the reported ACC and ASR in the original paper are 94.65% and 99.32%. The reason of such difference is that the poisoned samples are optimized together with model parameters in input-aware attack, and when the optimized poisoned dataset are used to train another model, the difference of model parameters may significantly downgrade the backdoor effect.
>
> - *Our defense:* As shown in the last two columns of Table 10, our defense methods significantly reduce ASR while maintaining high ACC, showing its effectiveness of defending against the attack. However, it is also notable that the ASR is higher than that of defending against other attacks. The main reason is that the attack capability of this poisoned training dataset is weak (*i.e.*, only 75.96% ASR), while most other attacks achieve as high as 95% ASR. Recall that our FCT metric is based on the sensitivity of poisoned samples to transformations, which is mainly due to the overfitting of the backdoored model to the trigger. Given a weak attack, the overfitting to the trigger may be alleviated, and thus weakens our defense to some extent. Although alleviating the overfitting to the trigger could somewhat weaken our defense, the attack significantly sacrifices its own attack performance. Seeking an attack which avoids the overfitting of the trigger and also has high attack performance may be a promising direction for the poisoning-based backdoor attack.
>
>   Anyway, the performance of our methods are still good. We infer that (a) there is no explicit or implicit optimization of trigger that encourages the features of poisoned samples insensitive to transformations, and (b) our methods are empirically proved robust even to adaptive attacks where the trigger is optimized to realize the aforementioned insensitivity (shown in the response to Reviewer FGt5: Q3).
>
> Besides, we have actually conducted several experiments to validate the effectiveness of our methods against attacks which have dynamic triggers. SSBA [9] is a strong attack which generates sample-specific triggers similar to those in input-aware attack. But different from the low ASR, SSBA could reach ASR as high as 99.64% on ImageNet. As shown in Table 5 of Appendix D, our method could effectively reduce ASR from 99.64% to 0.09% while keeping the ACC as high as 83.77%, which demonstrates the effectiveness of our method against dynamic triggers.

---

### Official Review · Reviewer_FGt5 · 2022-07-10

**Rating:** 7
**Confidence:** 4
**Soundness:** 3 good
**Presentation:** 4 excellent
**Contribution:** 3 good

**Summary:**

In this paper the authors: (a) Attribute the effectiveness of backdoor attacks to overfitting which they measure by proposing a metric called Feature Consistency towards Transformations. Using this metric they partition a given training dataset into potentially clean and potentially polluted samples and (b) They propose two defense algorithms (i) train a secure model from scratch (ii) Training a backdoored model first then removing the backdoor from the model.

To show (a) they pass the original image with the trigger and an augmentation of the triggered image to the same model and calculate the L2 distance in the feature embedding space. They define large distances mean more sensitivity. Using this metric they create a "sample-distinguishment" SD module to partition a given dataset by thresholding the FCT scores.

For defending against backdoor attacks from a potentially polluted dataset they propose two approaches:
1. Secure Training From Scratch: They follow an approach similar to [29] Contrastive Supervised Learning where they have a two term contrastive loss (a) self-supervised loss on clean and polluted samples (b) supervised loss on clean vs polluted designated by the SD module using FCT thresholding. After they train the feature extractor, they train a classifier on top by minimizing a two term cross entropy loss (i) increasing the likelihood of predicting clean samples as their correct class (i) decreasing the likelihood of predicting the identified polluted samples as incorrect.

2. Unlearn and relearn: on the found clean and polluted sets, they iteratively decrease the likelihood of the polluted predictions and increase the likelihood of the clean predictions using cross entropy loss.

They show with extensive ablation studies the effectiveness of the different choices they make in their defense algorithms.


**Questions:**

- Are the chosen hyper parameters easily transferrable to other datasets and network architectures?
- How sensitive is FCT to the dimensionality of embeddings since it's calculating a euclidean distance? Will it become harder to find a threshold for higher dimensions?
- How are the 0.05 and 0.2 thresholds set for the SD module?
- Is it possible to train a trigger in a whitebox setting with all of the hyper parameters of the defense method know to the attacker?



**Limitations:**

- An important aspect of the work is setting the threshold for SD. It is not explained how this threshold can be tuned without knowing beforehand which samples contain the trigger and the distribution of distances. Or it's not discussed how sensitive the method is setting these thresholds to a different value.

- A problem can be that in a whitebox setting the attacker can design the trigger such that FCT and its set thresholds are not effective by incorporating FCT in attack generation process by adding the inverse FCT to their optimization objectives for finding the trigger.

**Strengths And Weaknesses:**

Strengths:
- Novelty of their method and uncovering an interesting fact about the backdoored models and triggers.
- Presentation and clarity.
- Thorough experimentation for supporting their claims on the proposed defense method.

Weaknesses:
- All the experiments are on a single dataset, CIFAR. This raises questions around how easy it is to tune the different hyper parameters in the model on a different dataset and model architecture.

- More experiments missing about the detection rate of the FCT metric. Simple ROC curves could give a sense of the power of their metrics. I understand that the first ablation study is trying to indirectly address this but since this is such an important idea in the paper, it could be isolated and tested.

---

> ### Author Response · Authors · 2022-08-01
> **Response to FGt5: Q1**
>
> **Q1:** More experiments missing about the detection rate of the FCT metric and the sensitivity to hyper-parameters.
>
> **R1:** Before directly answering these two concerns, we would like to briefly list the related experiments we have conducted. Then, based on these existing experiments, we will give responses to the reviewer's concerns.
>
> **Related experiments we have conducted:**
>
> - **(1) First part of Section 4.3** answers questions as: How about the effectiveness of the SD module? How does our proposed FCT metric perform compared with other metrics?
>
> - **(2) Appendix E** answers questions as: How is the precision of the clean and poisoned samples distinguished by the FCT metric, respectively? How do the selected data transformations affect the precision?
>
> - **(3) Appendix F** answers questions as: How sensitive is our method to the proportion values $\alpha_c, \alpha_p$? How do they affect the defense performance, respectively?
>
> - **(4) Appendix G** answers questions as: How stable is our method against different poisoning rates, even with the invariant $\alpha_c, \alpha_p$? Is the SD module still effective?
>
> We think that Appendix E,F,G could answer most of your concerns with adequate details. For your convenience, here we reorganize the above experiments to present a more clear explanation, as follows.
>
> **(a) Detection rate of the FCT metric:** The detection rate you mentioned is exactly the precision we defined in Appendix E. Specifically, the clean(poison)-precision refers to precision of the distinguished clean(poisoned) samples $\hat{D}_c$($\hat{D}_p$). As shown in Figure 11, we presented the clean/poison-precision matrix under different pairs of transformations adopted in the FCT metric. It is shown that the precisions are very high (almost up to 100%) in most cases. Specifically, in our main experiments, we specified the first transformation as *rotate* and the second one as *affine*, corresponding to the precision at Row 1, Column 2 of each precision matrix, which is as high as 100% in most cases. We think that the study could demonstrate the effectiveness of the FCT metric, as well as its stableness to different transformations.
>
> **(b) Sensitivity to $\alpha_c, \alpha_p$?:**  $\alpha_c$($\alpha_p$) denotes the proportion of samples that are identified as clean(poisoned). As shown in Figure 3, the ground-truth clean samples tend to have a smaller FCT, while the ground-truth poisoned samples have a larger FCT. Thus, we sorted the training samples according to the ascending order of FCT, where the first $\alpha_c$ samples are identified as clean, while the last $\alpha_p$ as poisoned. More details of this algorithm are presented in Appendix A.1.
>
> Note that many existing backdoor works adopted the setting of 10% poisoning rate, which is also adopted in our main experiments (see Line 195). To avoid incorrectly identifying ground-truth clean samples as poisoned, we set $\alpha_p$ as 5% in the main experiments. Besides, we also conducted experiments in Appendix G to see how this $\alpha_p$=5% setting performs under an exact poisoning rate of 5%. Besides, we find that with $\alpha_p$=5%, correctly identifying a few clean samples (*i.e.*, $\alpha_c$=20%) is sufficient to achieve good defense performance using our methods, which is empirically validated by the following ablation study.
>
> In order to evaluate the sensitivity of our methods to different values of $\alpha_c$ and $\alpha_p$, we fixed one hyper-parameter while varying the other one to see the changing of the defense performance, as shown in Figure 13 in the Appendix F.
>
> - **Fixing $\alpha_p$ as 5% and varying $\alpha_c$ from 0% to 80%:** As shown in Figure 13(a), we found that as $\alpha_c$ grows, ACC increases steadily and finally converges. However, when $\alpha_c$ is too large (eg. 80%), the distinguished clean samples may contain ground-truth poisoned ones, resulting in the rise of ASR. Hence, the range from 20% and 40% is appropriate for $\alpha_c$.
>
> - **Fixing $\alpha_c$ as 20% and varying $\alpha_p$ from 0% to 20%:** As shown in Figure 13(b), we found that with the increase of $\alpha_p$, ASR declines steadily and finally converges. Nevertheless, excessive distinguished poisoned samples with larger $\alpha_p$ could hurt the model and lead to a reduction in ACC. Therefore, a moderate $\alpha_p$ (*e.g.*, the range from 5% to 10%) is preferred.
>
> The studies shown in Figure 13 demonstrate that our method could show superior and stable defense performance within a relatively wide range of values of $\alpha_c$ and $\alpha_p$.
>
> Moreover, in addition to the above analysis, we have also presented many other experiments in Appendix E,F,G to explore the SD module from different aspects.

---

> ### Author Response · Authors · 2022-08-01
> **Response to FGt5: Q2**
>
> **Q2:** All the experiments are on a single dataset. This raises questions around how easy it is to tune the hyper-parameters on a different dataset and model architecture, and how sensitive FCT is to the dimensionality of embeddings?
>
> **R2:** Thanks for this insightful comment. Firstly, we would like to clarify that we have actually conducted experiments on three datasets, including CIFAR-10, CIFAR-100 and an ImageNet subset, as demonstrated at Line 196-197. Due to space limit, the results on ImageNet are shown in Table 5 in Appendix D. In the following, we will explain the sensitivity of hyper-parameters to different datasets, model architectures, and feature dimensionalities, respectively.
>
> **Sensitivity to datasets:** As mentioned in Appendix C.5, we adopt the same setting of hyper-parameters on all of the datasets. Taken the D-BR method as an example, the average ASR reaches as low as 0.31%, 0.07% and 0.02% on the datasets. Meanwhile, compared with the backdoored model, the average ACC merely drops by 0.04%, 1.07% and 1.92%. The superior performance across three datasets demonstrates that the hyper-parameters are stable across different datasets.
>
> **Table 7: Performance under different model architectures against BadNet attack on CIFAR-10 dataset.**
>
> | Model architecture | Dimensionality of feature representation | Clean-precision of $\hat{D}_c$ | Poison-precision of $\hat{D}_p$ | Backdoored  model (ACC) | Backdoored model (ASR) | D-BR (ACC) | D-BR (ASR) | D-ST (ACC) | D-ST (ASR) |
> | ------------------ | ---------------------------------------- | ------------------------------ | ------------------------------- | ----------------------- | ---------------------- | ---------- | ---------- | ---------- | ---------- |
> | ResNet-18          | 512                                      | 100%                           | 100%                            | 91.64%                  | 100%                   | 92.83%     | 0.40%      | 92.77%     | 0.03%      |
> | ResNet-50          | 2048                                     | 100%                           | 9.00%                           | 90.88%                  | 100%                   | 88.47%     | 0.00%      | 90.32%     | 5.89%      |
> | VGG-19             | 512                                      | 100%                           | 100%                            | 91.09%                  | 100%                   | 90.90%     | 0.00%      | \          | \          |
> | DenseNet-161       | 8832                                     | 99.93%                         | 21.09%                          | 90.84%                  | 100%                   | 89.82%     | 0.00%      | \          | \          |
>
> **Sensitivity to model architectures and feature dimensionalities:** In addition to the ResNet-18 we have tested, here we conduct experiments on another three mainstream model architectures, including ResNet-50, VGG-19 and DenseNet-161, with the same parameter setting. Results are shown in Table 7. Besides, we uniformly choose the output of the penultimate layer as the feature representation, resulting in different dimensionalities. Thus, these results can also reflect the sensitivity to feature dimensionalities.
>
> In the following, we will analyze the effectiveness of different modules so as to evaluate the sensitivity of the hyper-parameters.
>
> - **Effectiveness of SD module:** Both ResNet-18 and VGG-19 achieve 100% precision. In contrast, the poison-precision of ResNet-50 and DenseNet-161 is relatively low, indicating that with the increase in the dimensionality, the gap between the FCT of clean and poisoned samples may be smaller. However, since their clean-precision is as high as about 100 % and that our methods are robust to wrong distinguishment as analyzed in Appendix E, this low poison-precision won't significantly influence the final performance which is analyzed subsequently.
> - **Effectiveness of BR module:** Compared with the performance of the backdoored model, our proposed BR module (*i.e.*, the D-BR method) could reduce ASR from 100% to 0% on the three new architectures. Meanwhile, ACC drops by 2.41% at most.
> - **Effectiveness of ST module:** Note that since SupContrast [29] (used in stage 1 of D-ST) only released codes for the ResNet architecture, and due to the time limit of rebuttal, here we didn't evaluate the ST module on VGG-19 and DenseNet-161. Compared with the backdoored model which directly employs the supervised learning, our proposed ST module (*i.e.*, the D-ST method) trains a secure model from scratch.
>
> **Summary:** The hyper-parameters of our methods are generalizable across datasets, model architectures and feature dimensionalities. However, we also found that the feature dimensionality may affect the setting, as the Euclidean distance used in our FCT metric may not be suitable for high dimensonal feature space. We will explore it in the future. We sincerely appreciate the reviewer's insightful comments once again.

---

> ### Author Response · Authors · 2022-08-01
> **Response to FGt5: First part of Q3**
>
> **Q3:** Adaptive attack: "Is it possible to train a trigger in a white-box setting with all of the hyper-parameters of the defense method known to the attacker?"
>
> **A3:** Thanks for this constructive suggestion. In the following, we firstly clarify the detailed setting, the objective function and the implemented algorithm of the adaptive attack. Then we will present the defense performance of our methods against this adaptive attack, and provide an analysis of the results.
>
> **What the adaptive attacker knows:** As suggested by the reviewer, here we adopt a full white-box setting that all of the hyper-parameters of the defense method are known to the attacker. Specifically, the attacker knows: **(a)** the defender will distinguish samples according to the sensitivity of samples to transformations which is measured by the FCT metric (*i.e.*, $\Delta_{trans}(x; \tau,f_{\theta_e}) = \Vert f_{\theta_e}(x)-f_{\theta_e}(\tau(x)) \Vert_2^2$), **(b)** $f_{\theta_e}(\cdot)$ represents the feature extractor of a backdoored model which is trained on the poisoned dataset with supervised learning, and **(c)** what kind of transformations $\tau(\cdot)$ and the model architecture the defender would use.
>
> **The objective function of the adaptive attack:** The goal of the adaptive attacker is to optimize a trigger which could make the feature representations of the poisoned samples insensitive to the transformations. Note that the feature representations are certain intermediate outputs of *a backdoored model*. To this end, the adaptive attack could be formulated as follows:
>
> $\min_{\theta_e, \theta_c, \delta} \frac{1}{\vert \bar{D_{train}}\vert}\sum_{(x,y)\in \bar{D_{train}}}-\log [h_{\theta_c}(f_{\theta_e}(x))]_y$
>
> $+ \frac{1}{2\vert D_p\vert} \sum_{(x,y)\in D_p} \Vert f_{\theta_e}(x)-f_{\theta_e}(\tau(x)\Vert_2^2,$
>
> where $\bar{D}_{train}=D_c \cup D_p$
>
> and $D_p = \{(x_i\oplus \delta, t)\}_{i=1}^{m_p}$. The first item is inserting backdoor into the model while the second item is realizing the insensitivity.
>
> **The algorithm to optimize the above objective function:** We adopt the patch-based type backdoor attack, *i.e.*, BadNets, and replace its static grid pattern trigger (shown in Figure 9(a) in Appendix) as a learnable variable. The trigger variable is initialized as the random noise sampled from uniform distribution. We optimize the parameters of the model and the trigger variable simultaneously, using the standard backpropagation with SGD, with the maximal 100 epoches.

---

> ### Author Response · Authors · 2022-08-01
> **Response to FGt5: Second part of Q3**
>
> **Table 8: The FCT mean and variance of clean and poisoned samples, clean-precision and poison-precision under different model-training epochs.**
>
> | num of training epochs $e_t$ | FCT Mean of clean samples | FCT Var of clean samples | FCT Mean of poisoned samples | FCT Var of poisoned samples | Clean-precision | Poison-precision |
> | ---------------------------- | ------------------------- | ------------------------ | ---------------------------- | --------------------------- | --------------- | ---------------- |
> | 2                            | 0.00                      | 0.00                     | 24.26                        | 22.56                       | 100%            | 100%             |
> | 100                          | 0.31                      | 7.01                     | 7.70                         | 23.07                       | 99.38%          | 81.36%           |
>
> **Table 9: Performance of the backdoored model and our proposed methods against the adaptive attack with trigger trained with different epochs.**
>
> | num of training epochs $e_t$ | Backdoored model (ACC) | Backdoored model (ASR) | D-BR (ACC) | D-BR (ASR) | D-ST (ACC) | D-ST (ASR) |
> | ---------------------------- | ---------------------- | ---------------------- | ---------- | ---------- | ---------- | ---------- |
> | 2                            | 90.53%                 | 99.98%                 | 91.36%     | 0.81%      | 93.51%     | 0.02%      |
> | 100                          | 90.70%                 | 99.99%                 | 88.51%     | 3.33%      | 88.82%     | 4.54%      |
>
> **Results and analysis**: The results of the SD module against the trained poisoned dataset of the adaptive attack are shown in Table 8, and the defense results of the proposed D-BR and D-ST are shown in Table 9. Here, we present two cases: the attacker with 2 training epochs and 100 training epochs.
>
> **(a) Defense against adaptive attack with 2 training epochs**: As described in Section C.5 in Appendix (see Line 527), it is notable that the backdoored model adopted to compute the FCT metric is trained with just $e_t=2$ epochs, since it is enough to insert the backdoor. Thus, we firstly suppose that the adaptive attacker also uses 2 epochs to train the trigger.
> As shown in the first row of Table 8, there is still remarkable difference on the distribution between the poisoned and clean samples, similar to what is shown in Figure 3 in the manuscript. Since figures cannot be shown here, we exhibit the distribution in the form of mean and variance. For clean samples, both the mean and variance of their FCT values are very close to 0.00. In contrast, for poisoned samples, the mean and variance are 24.26 and 22.56, respectively. More specifically, the smallest FCT value among all poisoned samples is 3.81, and there are only 20 clean samples (out of the total 45500 clean samples) that are greater than 3.81. The result demonstrates that the overlap between the distribution of clean and poisoned samples is tiny, which is helpful for distinguishing samples. As a result, both the clean-precision and the poison-precision (defined in Appendix E, *i.e.*, the precision of the distinguished clean and poisoned samples, respectively), are **100%**. It illustrates that our SD module is still effective in distinguishing samples, not influenced by the optimized trigger.
>
> Consequently, as shown in Table 9, when we apply the BR module based on the above distinguishing result, our proposed D-BR method could effectively reduce ASR from 99.98% to 0.81%, and even improve ACC by 0.83%, compared to the backdoored model. Moreover, the proposed D-ST method could even perform better. It securely trains a model from scratch which has ACC as high as 93.51% (2.98% higher than that of the backdoored model) and keeps ASR as low as 0.02%. The defense performance demonstrates that our proposed two defense methods are still effective in defending against the adaptive attack.
>
> The main reason is that the backdoor could be quickly learned within 2 epochs (*i.e.*, the value of the first loss term in the above objective function decreases quickly), while the regularization term in the objective is difficult to be fitted within 2 epochs. Consequently, the poisoned samples with the trigger trained in 2 epochs are still more sensitive to transformations than clean samples, causing the good defense performance of our methods.

---

> ### Author Response · Authors · 2022-08-01
> **Response to FGt5: Third part of Q3**
>
> **(b) Defense against adaptive attack with 100 training epochs**: As analyzed above, since the regularization term in the attack's objective function is not well fitted in 2 epochs, we then suppose the adaptive attacker trains the trigger with 100 epochs. However, as shown in the last column of Table 8, we find out that there is still a gap between the distribution of clean and poisoned samples since the mean FCT value of clean samples is 0.31, which is much smaller than that of poisoned sample, *i.e.*, 7.01. However, the overlap between the clean and poisoned distributions becomes larger, compared to the above case of 2 epochs. Consequently, the poison-precision decreases to 81.36%, while the clean-precision is still very high, up to 99.38%. It illustrates that our SD module is somewhat affected by the optimized trigger.
>
> As shown in Table 9, when we apply the BR module and the ST module subsequently, our proposed defense methods are still effective. For the backdoor removal paradigm, compared to the backdoored model, although ACC slightly drops by 2.19%, D-BR could effectively reduce ASR from 99.99% to 3.33%. For the secure training paradigm, D-ST also achieves performance comparable to D-BR. These results demonstrate that compared to 2 epochs, the attack with the trigger trained with 100 epochs is more threatening. But our proposed methods could still obtain a high-performance model with relatively low ASR, illustrating that it would not be easy for the adaptive attack to insert a backdoor deep into model, even with the white-box setting.
>
> The possible reason is that although the trigger trained with 100 epochs could better encourage the insensitivity of poisoned samples in the corresponding backdoored model, our SD module will not use this backdoored model. Instead, our SD module will train a new backdoored model with only 2 epochs based on the returned poisoned datasets. Consequently, the sensitivity of poisoned samples becomes larger in the new backdoored model used in SD module, causing the failure of the adapative attack.
>
> **Summary:** According to above evaluations and analysis, we surmise that, since the speed of inserting the backdoor and that of training a good adaptive trigger are much different, it is difficult to obtain a good trigger and a model similar to the defender-used one at the same time, causing the difficulty in designing a successful adaptive attack. In this sense, we could claim that to some extent, our defense method is robust to the possible adaptive attack, even in the most restricted white-box setting.
> Furthermore, in practice, even with a potential successful adaptive attack, simply changing the type of transformations or the model architechture used in our defense methods could be considered as an easy way to defend against the adaptive attack.
> However, we will keep exploration of possible adaptive attacks against our proposed defense methods in our future work, to continually contribute to the development of the backdoor learning.

---

> ### Comment · Reviewer_FGt5 · 2022-08-09
> **Thanks for the detailed responses**
>
> I acknowledge I read all the responses and I want to thank the authors for their thorough explanations. My scores stay the same.
>
> I would like to see the limitation on the feature dimensionality in the final text of the paper.

---

> > ### Author Response · Authors · 2022-08-09
> > **Thanks for your encouraging comments**
> >
> > Dear Reviewer,
> >
> > Thanks for you constructive comments which are really beneficial to our work. And we will add the limitation on the feature dimensionality into the final manuscript.

---

### Official Review · Reviewer_dNMX · 2022-07-17

**Rating:** 8
**Confidence:** 3
**Soundness:** 4 excellent
**Presentation:** 4 excellent
**Contribution:** 4 excellent

**Summary:**

In this paper, the authors reveal the sensitivity of poisoned samples to transformations and propose a sensitivity metric FCT. And the authors  propose two effective backdoor defense methods for training a secure model from scratch and removing backdoor from the backdoored mode.

**Questions:**

1. In the introduction, it is suggested that the authors add the background to the application of the proposed method, thus enhancing the motivation for the paper.
2. The authors only list the researches for the image super-resolution and don't detail the reasons why these approaches are not sufficient for their goal.

**Limitations:**

Yes,  the authors adequately addressed the limitations and potential negative societal impact of their work.

**Strengths And Weaknesses:**

The authors  design a simple sensitivity metric to distinguish poisoned samples from clean samples in the untrustworthy training set, and  propose two effective backdoor defense methods. The paper tackles an interesting issue, and the efforts of the authors are clear in investigating the problem and in writing the manuscript.

---

> ### Author Response · Authors · 2022-08-01
> **Response to dNMX**
>
> **Q1:** In the introduction, it is suggested that the authors add the background to the application of the proposed method, thus enhancing the motivation for the paper.
>
> **A1:** Thanks for pointing out the importance of the threat model we discuss in this paper, and we are encouraged by your positive comments on our work. Following your suggestion, we will add the following background into the revised manuscript, as follows.
>
> The tremendous success of DNNs in a variety of fields relies heavily on the growing availability of large datasets. Training a DNN with good performance often requires a large amount of training data. In this way, annotating all data manually is considered to be time-consuming and unrealistic. As a result, practitioners usually obtain data by purchasing from a third-party data provider or collecting some open-sourced databases. However, some malicious attackers may poison the dataset so that when a user downloads this dataset and trains a model on it locally, the trained model could contain certain stealthy backdoor, which could be activated by a particular trigger by the attacker. This kind of attack is called data poisoning-based backdoor attack, which could pose a serious security threat to the DNN training. Therefore, designing a method to defend against this kind of attack is of great value and importance, which is the motivation for our work.
>
> ------
>
> **Q2:** The authors only list the researches for the image super-resolution and don't detail the reasons why these approaches are not sufficient for their goal.
>
> **A2:** We are confused about the key word *image super-resolution* in the above concern, as our work is not related to this topic. Would you please provide some clarification? We are willing to answer any further concern.

---

### Meta-Review · Area_Chair_vk2f · 2022-08-25

**Recommendation:** Accept
**Confidence:** Certain

**Metareview:**

The authors propose a new method for defending against backdoor attacks which is based on the observation that poisoned samples are more sensitive to transformations than clean samples. They design a metric called \textit{feature consistency towards transformations (FCT)} to distinguish poisoned samples from clean samples in the untrustworthy training set.

The paper received favorable reviews and has made substantial updates during the rebuttal phase to the general satisfaction of the reviewers. I thus recommend accept.


**Award:**

No

---

### Decision · Program_Chairs · 2022-09-14

Accept